# Epithelial disruption drives mesendoderm differentiation in human pluripotent stem cells by enabling TGF-β protein sensing

Thomas Legier [1,3], Diane Rattier[1,3], Jack Llewellyn [1], Thomas Vannier [1], Benoit Sorre [2], Flavio Maina [1] & Rosanna Dono [1] ✉

The processes of primitive streak formation and fate specification in the mammalian epiblast rely on complex interactions between morphogens and tissue organization. Little is known about how these instructive cues functionally interact to regulate gastrulation. We interrogated the interplay between tissue organization and morphogens by using human induced pluripotent stem cells (hiPSCs) downregulated for the morphogen regulator GLYPICAN-4, in which defects in tight junctions result in areas of disrupted epithelial integrity. Remarkably, this phenotype does not affect hiPSC stemness, but impacts on cell fate acquisition. Strikingly, cells within disrupted areas become competent to perceive the gastrulation signals BMP4 and ACTIVIN A, an in vitro surrogate for NODAL, and thus differentiate into mesendoderm. Yet, disruption of epithelial integrity sustains activation of BMP4 and ACTIVIN A downstream effectors and correlates with enhanced hiPSC endoderm/mesoderm differentiation. Altogether, our results disclose epithelial integrity as a key determinant of TGF-β activity and highlight an additional mechanism guiding morphogen sensing and spatial cell fate change within an epithelium.

Gastrulation is a major morphogenetic process in animal development during which the embryonic germ layers are generated and the basic body plan is laid down. In mammals, gastrulation is first marked by the appearance of the primitive streak (PS), a transient structure arising by thickening of the epiblast. Epiblast cells recruited and ingressing into the PS will acquire a mesendoderm (MES) fate, marked by co-expression of the transcription factors T/Brachyury (BRACH) and Eomesodermin (EOMES). Subsequently cells undergo epithelial-mesenchymal transition (EMT) to initiate differentiation into the mesoderm (ME) and definitive endoderm (DE) germ layers[1,2]. Instead, epiblast cells that do not ingress through the streak will become the future ectodermal layer.

A mechanistic understanding of how mammalian gastrulation is initiated and how pluripotent cells of the epiblast acquire different fates has attracted considerable scientific interest given the impact that findings can have for developmental biology and for clinical application. Great insights have come from genetic and embryological studies carried out in the mouse embryo. These studies have led to the proposal that gastrulation is initiated by means of biochemical signals like WNT, the TGF-β superfamily members, BMP and NODAL, and their inhibitors, which establish a signaling crosstalk between embryonic and extra-embryonic cells[2]. In this scenario, gastrulation is initiated when epiblast cells are instructed by BMP4, secreted by the extra embryonic ectoderm, to produce the secreted signaling protein WNT3[2,3]. WNT3 in turn stimulates epiblast cells to express and release NODAL that will feed back to the extraembryonic ectoderm to maintain BMP signaling in this tissue as well as the overall signaling crosstalk between the two cellular compartments[2–5]. As a result of this signaling

[1]Aix Marseille Univ, CNRS, IBDM, Turing Center for Living Systems, NeuroMarseille, Marseille, France. [2]Institut Curie, Université PSL, Sorbonne Université, CNRS UMR168, Laboratoire Physico Chimie Curie, Paris, France. [3]These authors contributed equally: Thomas Legier, Diane Rattier. ✉e-mail: rosanna.dono@univ-amu.fr

crosstalk between BMP, WNT and NODAL, the PS is formed and the epiblast becomes patterned.

Recently, this configuration has been challenged by studies showing that these signaling interactions may be further modulated by topological cues and by the epithelial organization of the epiblast. In particular, the use of sophisticated live imaging tools in mouse embryos has highlighted that BMP receptors localize at the basolateral membrane of mouse epiblast facing a narrow interstitial space located between the epiblast cells and the underlying visceral endoderm. Interestingly, the lack of tight junctions (TJs) between the extra-embryonic ectoderm and the edge of the epiblast at the posterior embryo site enables the diffusion of BMP4 through this interstitial space, allowing ligands to reach basolateral localized receptors and initiate gastrulation[6].

As these mechanical studies are arduous in embryos given the small size and inaccessibility of early post-implantation embryos, human pluripotent stem cells (hPSCs) have become a reference system to dissect physical processes and feedback interactions with bio-chemical signals. Pioneering studies with hPSCs grown on a disk-shaped plate, to mimic the mechanical constrains of the human epi-blast (called '2D gastruloids' or 'micropattern' in the following), have highlighted that physical cues, such as colony size, shape and cell density, can impact on the rate and trajectory of hPSC differentiation by acting on the balance between differentiation-inducing and -inhi-biting factors[7,8]. Subsequent experiments in which the 2D gastruloid technology was applied to the study of tissue patterning have revealed a remarkable capability of hPSCs to recapitulate aspects of germ layer patterning, such as simultaneous formation of the radially organized embryonic germ layers when exposed to the gastrulation-initiating signal, BMP4[9,10]. Of note, additional mechanistic studies have revealed that this in vitro patterning process relies on the epithelial architecture of the hPSC layer, which determines accessibility of BMP4 receptors in cells and formation of morphogen gradients[11,12]. By using this con-finement strategy, it has also been proposed that regions of high cell–cell tension in the hPSC layer foster the appearance of gastrulation-like nodes (PS-like) and mesoderm differentiation in response to BMP4 stimulation[13]. This is due to the release of β-catenin from cell–cell adhesion sites and its nuclear translocation. Collectively, these results raise the possibility that intrinsic changes in the epiblast cell layer could trigger the onset of gastrulation and MES differentia-tion by sensitizing cells to morphogens.

The heparan sulphate proteoglycan GLYPICAN-4 (GPC4)[14], is a key regulator of signals controlling the balance between PSC self-renewal and differentiation (e.g. BMP, WNT, FGF)[15–20]. By performing functional analysis of human induced PSCs (hiPSCs) with reduced GPC4 protein levels (GPC4sh hiPSCs), we discovered that these cells provide a rele-vant biological system to obtain additional insights into the interplay between the physical properties of epiblast cells and biochemical signals during gastrulation.

Here we show that in contrast to control cells, GPC4sh hiPSCs display altered epithelial integrity with areas of abnormal TJ distribu-tion. Of note, GPC4sh hiPSCs display enhanced differentiation poten-tial into MES, ME and DE lineages, under classical differentiation protocols compared to control cells. Strikingly, areas of disrupted epithelial integrity correlate precisely with expression of the MES markers BRACH and EOMES, indicating that the morphological phe-notype of the GPC4sh hiPSC layer sensitizes cells to respond to mor-phogens. By performing stimulation assays with BMP4, ACTIVIN A and WNT3A ligands, we show that epithelial integrity regulates the ability of hiPSCs to respond to BMP4 and ACTIVIN A but not WNT3A. In addition, we demonstrate that disruption of epithelial integrity sus-tains activation of the BMP4 and ACTIVIN A signaling pathway over time. Thus, our findings highlight alteration of epithelial integrity as an additional mechanism for controlling BMP4 and ACTIVIN A sig-naling and differentiation processes.

## Results

### Loss-of-GPC4 affects hiPSC epithelial integrity by altering TJs

To investigate the consequences of disrupting epithelial organization following GPC4 downregulation in hiPSCs, we used two previously published hiPSC lines in which GPC4 was downregulated in the parental hiPSC 029 by means of shRNA-mediated GPC4 targeting. Briefly, we transduced the hiPSC line 029 with lentiviruses encoding two selected shRNAs named GPC4sh5 and GPC4sh2 to generate stable clones from each targeting sequence[14]. Among them, we selected GPC4sh5-c10 and the GPC4sh2-c3 clones as representative cells carrying the GPC4sh5 and the GPC4sh2 sequences, respectively (Fig. 1). The 029 hiPSC line transduced with the non-mammalian sequence GFP was used as a control (CTRLsh)[14]. RT-qPCR analysis of GPC4 expression showed no significant differences in GPC4 transcript levels between wild type (WT) and CTRLsh hiPSCs (Fig. 1a) as previously reported[14]. In contrast, GPC4 transcript levels were reduced by 60%±14% in GPC4sh5-c10 and by 70% ±4% in GPC4sh2-c3 hiPSCs (Fig. 1a). Consistently, immunocytochemical analysis revealed that GPC4sh5 and GPC4sh2 cultures display reduced GPC4 levels compared to WT and CTRLsh (Fig. 1b).

Next, we evaluated the morphology and organization of controls and GPC4sh hiPSC cultures by analyzing epithelial features. In parti-cular, cells grown at high density (e.g. $1.5 \times 10^5$ cells/cm$^2$) were immu-nostained for the TJ proteins, Zona occludens 1 (ZO1) and Occludin (OCLN), by immunocytochemistry. Intriguingly, we found that GPC4sh cultures display areas with disrupted TJ organization, whereas control cultures form the classical epithelial sheet with organized and defined TJs (Fig. 1c and Supplementary Fig. 1a). Quantification revealed that this phenotype covers 39.1% ± 4% and 58.2% ± 11% of the cell layer in GPC4sh5-c10 and GPC4sh2-c3 hiPSCs, respectively, whereas TJ dis-rupted areas account only for 8.4% ± 3% and 7.4% ± 2% of the epithelial sheet in control hiPSCs (Fig. 1c, d). Rescue experiments demonstrated that epithelial integrity is restored in GPC4sh cultures following upregulation of GPC4 levels, thus showing that disruption of the epi-thelial integrity is due to GPC4 downregulation (Supplementary Fig. 1b, c). HiPSC cultures with downregulated GPC4 exhibit abnormal TJ organization even when cultured at low density ($0.2 \times 10^5$ cells/cm$^2$ and $1 \times 10^5$ cells/cm$^2$ instead of $1.5 \times 10^5$ cells/cm$^2$; Supplementary Fig. 1d). Finally, disruption of TJ organization in GPC4sh hiPSCs occurs in cells plated on different extracellular matrices (ECM) such as Matrigel (a mixture of Collagen, Laminin and growth factors), Laminin or Vitro-nectin (Supplementary Fig. 1e). This finding suggests that disruption of TJ organization in GPC4sh hiPSCs occurs also in the presence of more stringent and defined cultured conditions.

To assess the effect of GPC4 downregulation in a different hiPSC line, we examined TJ organization in the previously published AICS-0023 hiPSCs targeted with CTRLsh and GPC4sh (Supplementary Fig. 1f)[14]. Analysis of ZO1 distribution revealed extensive areas of dis-rupted TJ organization in the epithelial sheet of the AICS-0023 GPC4sh compared with the CTRLsh line (Supplementary Fig. 1g). This result highlights that disrupted TJ organization is triggered by GPC4 down-regulation and is not limited to one hiPSC line.

It is well known that loss of TJs occurs when epithelial cells switch to a mesenchymal state[21,22]. Therefore, we next examined the expres-sion of genes regulating epithelial and mesenchymal states in controls and GPC4sh hiPSCs using RNA-sequencing (RNA-seq) data. By com-paring transcript levels of 62 epithelial and 71 mesenchymal markers we did not find major differences between WT, CTRLsh and GPC4sh5 hiPSCs (Fig. 1e and Supplementary Table 1). These results were cor-roborated by western-blot analyses of some epithelial (ZO1, OCLN, E-Cadherin (E-CAD) and β-Catenin (β-CAT)) and mesenchymal (N-Cadherin (N-CAD)) markers (Fig. 1f). Consistently with our RNA-seq data, we found similar levels of ZO1, OCLN, E-CAD, β-CAT and N-CAD proteins in controls and GPC4sh hiPSCs (Fig. 1f). Overall, this analysis indicates that downregulation of GPC4 in hiPSCs does not pro-mote an EMT.

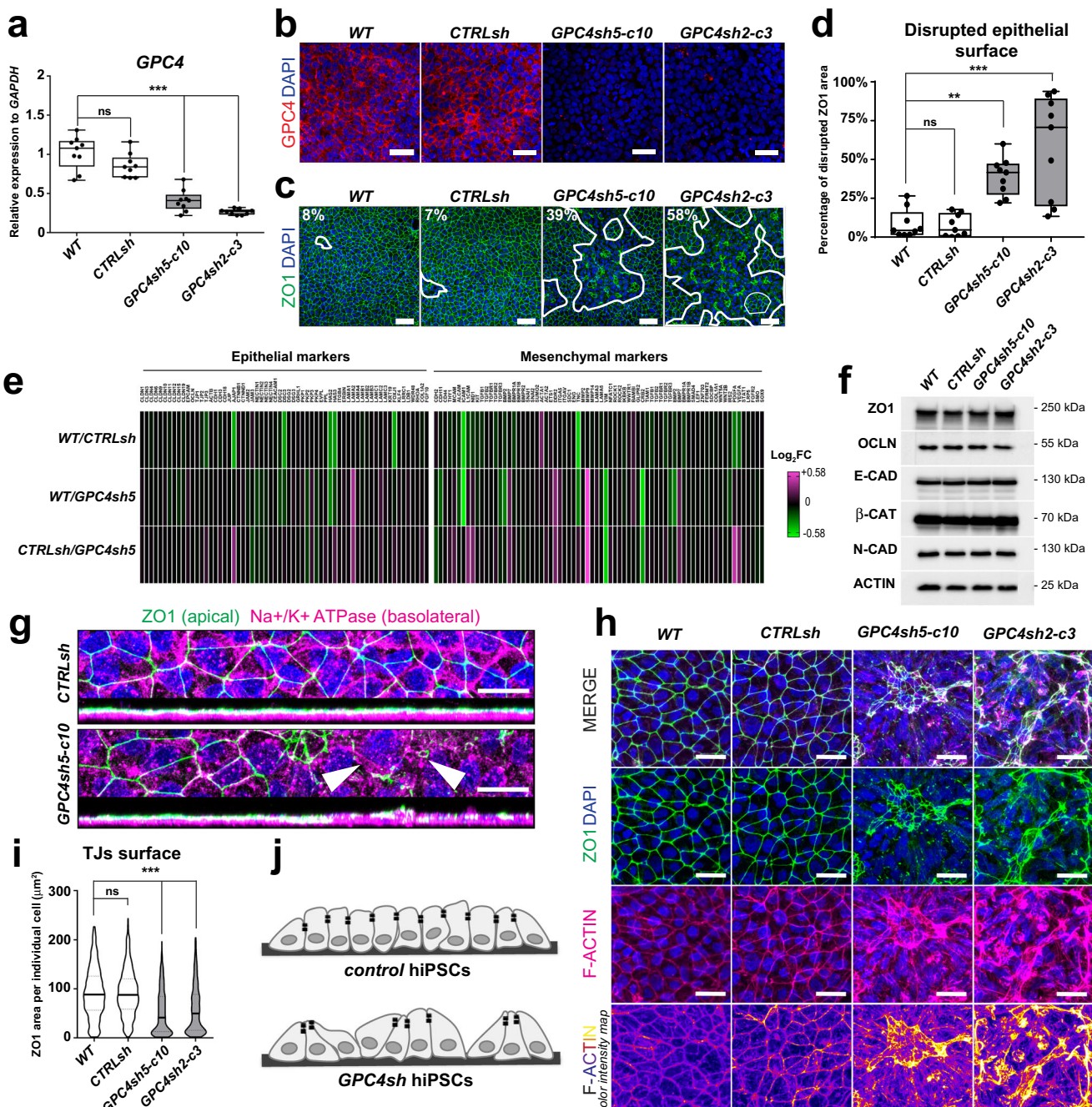

**Fig. 1 | GPC4 downregulation in hiPSC affects epithelial integrity by altering TJ distribution. a** RT-qPCR analyses of *GPC4* transcript levels in WT, CTRLsh, GPC4sh5-c10 and GPC4sh2-c3 029 hiPSCs. Box plots represent the median with interquartile range, the whiskers indicate min and max values, *n* = 9.
**b** Immunofluorescence analysis of GPC4 (red) and DAPI positive nuclei (blue) in WT, CTRLsh, GPC4sh5-c10 and GPC4sh2-c3 hiPSCs, *n* = 3. Scale bar: 50 μm.
**c** Immunofluorescence analysis of ZO1 (green) in WT, CTRLsh, GPC4sh5-c10 and GPC4sh2-c3 hiPSCs. Numbers indicate the percentage of disrupted areas. White outlines highlight zones of disturbed TJs, *n* = 9. Scale bar: 50 μm. **d** The extent of TJ disruption was quantified from staining shown in **c**. Box plots represent the median with interquartile range, the whiskers indicate min and max values, *n* = 9. **e** RNA-seq analyses of transcript levels of epithelial and mesenchymal genes in WT, CTRLsh, GPC4sh5-c10 hiPSCs. Data are represented as a heatmap of Log₂FC, *n* = 3. **f** Western-blot analysis of ZO1, OCLN, E-CAD, β-CAT and N-CAD in WT, CTRLsh, GPC4sh5-c10 and GPC4sh2-c3 hiPSC total protein extracts. ACTIN protein levels were used as

loading control, *n* = 3. Blots were processed in parallel. **g** Immunofluorescence analysis of ZO1 (green) and Na⁺/K⁺ ATPase (magenta), in CTRLsh and GPC4sh5-c10 hiPSCs. Pictures are presented as top and lateral view of stained cells. Arrowheads point to areas of abnormal apical-basal polarity, *n* = 3. Scale bar: 20 μm.
**h** Immunofluorescence analysis of ZO1 (green), F-ACTIN (magenta) in WT, CTRLsh, GPC4sh5-c10 and GPC4sh2-c3 hiPSCs, *n* = 10. Scale bar: 25 μm. **i** The surface delimited by the ZO1 protein in each individual cell was quantified from staining shown in **h**. Data are represented as violin plots (distribution around the median), *n* = 4000 cells per hiPSC line. **j** Schematic representation of epithelial cell morphology of control hiPSCs, and of disrupted TJs in GPC4sh hiPSCs. Statistical analysis for the overall figure: (**a**, **d**, **i**) one-way ANOVA followed by Dunnett's multiple comparison test. *P*-values: (***) <0.001, (**) <0.01, (*) <0.05, ns not significant. For all panels "*n*" corresponds to the number of biological replicates. Source data are provided as a Source Data file.

We then asked whether downregulation of GPC4 disrupts other aspects of the apical-basal polarity of epithelial cells. We examined the distribution in the Z-axis of baso-lateral (Na+/K+ ATPase and E-CAD) and apical (ZO1) marker proteins[23–25]. Immunocytochemical analyses of control hiPSC cultures showed that Na+/K+ ATPase and E-CAD maintained their basolateral localization below TJs (ZO1 immunostaining; Fig. 1g and Supplementary Fig. 1h, i). Instead, Na+/K+ ATPase or E-CAD staining were found intermingled with ZO1 in GPC4sh hiPSC cultures in areas with disrupted TJ organization (Fig. 1g and Supplementary Fig. 1h, i). These results highlight disruption of the apical-basal cell polarity in GPC4sh hiPSCs, which can lead to an apical exposure of the basolateral localized proteins.

To further characterize the changes in epithelial morphology of GPC4sh hiPSCs, we analyzed the state of TJs in conjunction with that of F-ACTIN cortex. In control hiPSC cultures, we observed that the F-ACTIN cortex is clearly defined and organized in a regular pattern matching the TJ localization (Fig. 1h), reflecting an equilibrated tissue tension[26]. Instead, in GPC4sh cultures we noticed that, within the areas of altered TJs, the F-ACTIN cortex is poorly defined and exhibits multiple foci of accumulation (Fig. 1h), indicative of high tissue tension[26]. Quantification of the apical cell surface size revealed that this surface is strongly reduced in GPC4sh cultures compared to control hiPSCs (Fig. 1i, j).

Taken together, these results show that downregulation of GPC4 in hiPSCs alters TJ distribution and apical-basal cell polarity. Although this phenotype does not affect epithelial cell identity, it disrupts the morphology and integrity of the hiPSC epithelial layer and enables apical exposure of basolateral localized proteins.

## Epithelial disruption does not affect hiPSC stemness

We next asked whether disruption of the epithelial integrity in hiPSC impacts on their stemness and differentiation properties. We found that GPC4sh hiPSCs stained as intensely for the stemness marker alkaline phosphatase as controls (Supplementary Fig. 2a), and that the expression levels of the epiblast marker OTX2 and of the core pluripotency genes OCT4, NANOG and SOX2 were not affected (Supplementary Fig. 2b–d). Similarly, no differences were found in the mitotic (phospho-Histone H3 positive cells) and apoptotic rates (cleaved-Caspase3 positive cells; Supplementary Fig. 2f–h) as well as in the cell cycle progression (Supplementary Fig. 2i). These results are consistent with data we reported previously[14]. To extend this analysis further, we used our RNA-seq data to follow the expression levels of 48 markers of PSCs in WT, CTRLsh and GPC4sh5 cultures maintained in self-renewal conditions. As above, the transcript levels of all PSC markers analyzed were not significantly changed between WT, CTRLsh and GPC4sh5 hiPSCs (Fig. 2a; Supplementary Table 2). Collectively, these findings show that disruption of epithelial integrity in hiPSCs does not affect stemness identity when cells are cultured in self-renewal conditions.

Previous studies have shown that the balance between self-renewal and differentiation of PSCs is coordinated by the crosstalk between the PI3K/AKT, RAF/MEK/ERK, and WNT/GSK3b signaling pathways that together impact on the ability of ACTIVIN A/SMAD2,3 to promote either self-renewal or differentiation into MES cells[27–31]. Therefore, we determined whether disruption of hiPSC epithelial integrity impacts on this regulatory signaling network. Western-blot analysis showed similar activation levels of AKT, ERK, GSK3a/b and SMAD2 pathways in control and GPC4sh hiPSCs maintained in self-renewal conditions (Fig. 2b). The antibody specificity was tested by using specific inhibitors of phosphorylation (Supplementary Fig. 2e). Consistently, RNA-seq analysis revealed that genes regulating fate acquisition such as GSC, BRACH, EOMES, FOXC1, FOXA1 and SOX1 were not expressed neither in WT and CTRLsh hiPSCs nor in GPC4sh5 cells (see Data availability). Together these results show that disruption of hiPSC epithelial integrity does not impact on

self-renewal and pluripotency of hiPSCs, nor primes them to express cell lineage markers when cells are cultured under stemness conditions.

## Enhanced mesendoderm fate acquisition in GPC4sh cultures

Recent studies have shown that differentiation of hiPSCs towards the MES lineage and into its DE and ME derivatives relies on hiPSC mechanics and epithelial organization properties[9,11,13,32]. Given the peculiar morphological phenotype of GPC4sh hiPSCs, we reasoned that GPC4sh cultures could be a powerful system to study the impact of epithelial integrity on hiPSC differentiation along MES, DE and ME embryonic lineages. To trigger hiPSC differentiation along these lineages we exposed cells to BMP4 and ACTIVIN A (Fig. 2c), a commonly used surrogate to activate aspects of the NODAL signaling pathway in vitro[33–37]. We then evaluated the differentiation profile of control and GPC4sh hiPSCs by using RT-qPCR analysis following lineage-specific markers.

In agreement with published studies, the MES markers EOMES, GSC, NODAL and MIXL1 were already expressed at day 1 (d1) of differentiation in control cells exposed to ACTIVIN A, and transcript levels were gradually downregulated from d1 to d5 (Fig. 2d and Supplementary Fig. 3a, b). When cells were kept in the presence of ACTIVIN A, expression of the DE markers SOX17, FOXA2, CER1 and CXCR4 started at d1, peaked between d2 to d3 and gradually reduced from d3 to d5 (Fig. 2e and Supplementary Fig. 3a, b)[33,38,39]. Instead, when ACTIVIN A was replaced by BMP4 at d2, the early ME markers MESP1 and TBX6 were gradually expressed from d2 to d4 then downregulated from d4/d5 (Fig. 2e and Supplementary Fig. 3a, b). At these late time points of differentiation, we observed concomitant expression of late ME markers, such as VEGFR2 and PDGFRa[34,40] (Fig. 2e and Supplementary Fig. 3a, b). Remarkably, by comparing the differentiation profile of GPC4sh hiPSCs versus control, we observed a 5- to 9-fold increase in transcript levels of all MES markers in GPC4sh hiPSCs (Fig. 2d and Supplementary Fig. 3a, b). Consistently, the expression of DE and ME markers also increased 5- to 12-fold and 5- to 19-fold, respectively, in differentiating GPC4sh cells compared to controls (Fig. 2e and Supplementary Fig. 3a, b). Thus, these results indicate that GPC4sh hiPSCs acquire enhanced differentiation potential towards MES and its derivatives lineages, DE and ME, than control hiPSCs. Consistent with RT-qPCR results, immunocytochemistry revealed a ~2.5 times higher percentage of cells expressing the early MES markers BRACH and EOMES at day 1 of differentiation in GPC4sh compared to control cultures (Fig. 2f, g and Supplementary Fig. 4a, b, e). Along the same line, the percentage of cells expressing the DE marker SOX17 and the ME marker PDGFRa at day 3 of differentiation was, respectively, ~17 times and >3 times higher in GPC4sh than control cultures (Fig. 2f, g and Supplementary Fig. 4c–e). Differentiation studies performed on the second hiPSC line, AICS-0023, showed a similar increase into BRACH and SOX17 positive cells in GPC4sh AICS-0023 cells compared to controls (Supplementary Fig. 5a, b), thus showing that this phenotype is not restricted to one hiPSC line.

As DE and ME fates acquisitions are accompanied by an EMT[21,22], we next evaluated the capabilities of GPC4sh cells to undergo an EMT fate switch during differentiation. Marker analysis revealed that GPC4sh hiPSCs undergo a more efficient EMT transition compared to control hiPSCs, as highlighted by the increased expression of N-CAD, SNAI1, SNAI2, TWIST1 and VIMENTIN mesenchymal markers, and by a more efficient switch in E-CAD to N-CAD expression (Fig. 3a). This E-CAD to N-CAD transition was also marked by the concomitant appearance of SOX17 and PDGFRa proteins in N-CAD positive cells (Fig. 3b, c). Altogether these results show that downregulation of GPC4 enhances hiPSC differentiation efficiency into MES cells as well as into their DE and ME derivatives with a concomitant EMT.

No significant changes in the percentage of mitotic and apoptotic cells were observed in control and GPC4sh hiPSCs (Supplementary

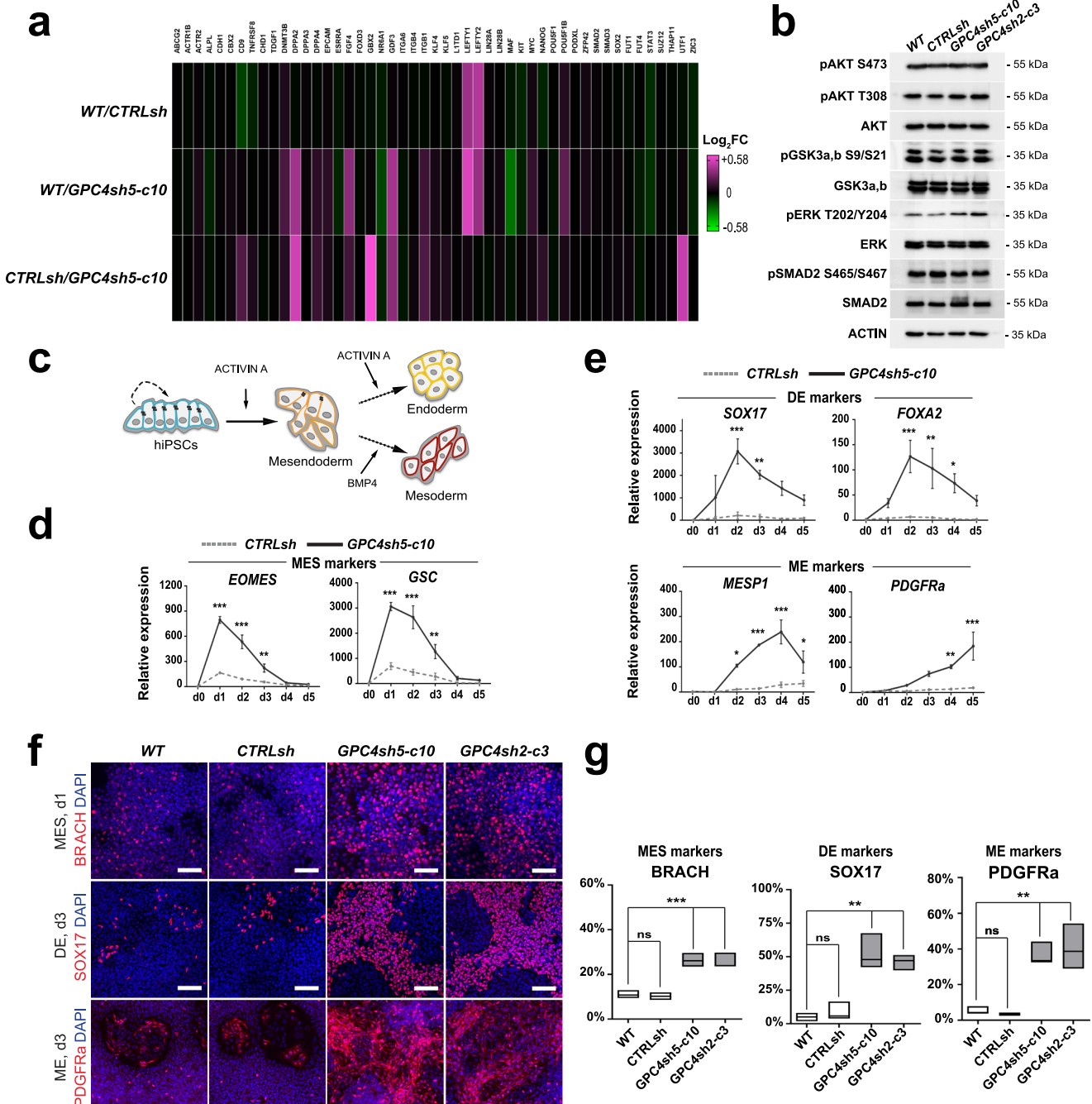

**Fig. 2 | Disruption of epithelial integrity does not affect stemness properties of hiPSCs but enhances differentiation potential. a** Transcript levels of genes regulating pluripotency were analyzed by RNA-seq in WT, CTRLsh and GPC4sh5-c10 029 hiPSCs at the undifferentiated stage. Data are represented as a heatmap of $Log_2FC$, $n = 3$. **b** Total protein extract of WT, CTRLsh, GPC4sh5-c10 and GPC4sh2-c3 029 hiPSCs at the undifferentiated stage were analyzed by Western-blot with pAKT S473, pAKT T308, AKT, pERK T202/Y204, ERK, pGSK3a, b S9/S21, GSK3a, b, pSMAD2 S465/S467 and SMAD2 antibodies. Note similar expression and phosphorylation levels of AKT, ERK, GSK3a, b and SMAD2 proteins in all hiPSC lines. ACTIN protein levels were used as loading control, $n = 3$. Blots were processed in parallel. **c** Scheme depicting the strategy of differentiation into MES and subsequently into DE and ME used. **d**, **e** CTRLsh and GPC4sh5-c10 029 hiPSCs were differentiated for 5 days into DE or ME and transcript levels of MES, DE or ME specific markers were analyzed by RT-qPCR. **d** MES markers (*EOMES* and *GSC*),

(**e**) DE markers (*SOX17* and *FOXA2*), (**e**) ME markers (*MESP1* and *PDGFRa*). Data were normalized to the d0 of CTRLsh and represented as mean ± SEM, $n = 3$.
**f** Immunofluorescence analysis of BRACH (red, MES), SOX17 (red, DE) or PDGFRa (red, ME) on WT, CTRLsh, GPC4sh5-c10 and GPC4sh2-c3 029 hiPSCs differentiated for 1 day into MES, 3 days into DE or 3 days into ME, $n = 3$. Scale bar: 100 µm.
**g** Percentages of BRACH, SOX17 and PGFDRa positive cells in WT, CTRLsh, GPC4sh5-c10 and GPC4sh2-c3 029 hiPSCs were quantified from staining shown in **f**. Box plots represent the median with min and max values, $n = 3$. Statistical analysis for the overall figure: (**d**, **e**) two-way ANOVA followed by Sidak's multiple comparison test, (**g**) two-way ANOVA, followed by Dunnett's multiple comparison test. *P*-values: (***) <0.001, (**) <0.01, (*) <0.05, ns not significant. For all panels "*n*" corresponds to the number of biological replicates unless stated otherwise. Source data are provided as a Source Data file.

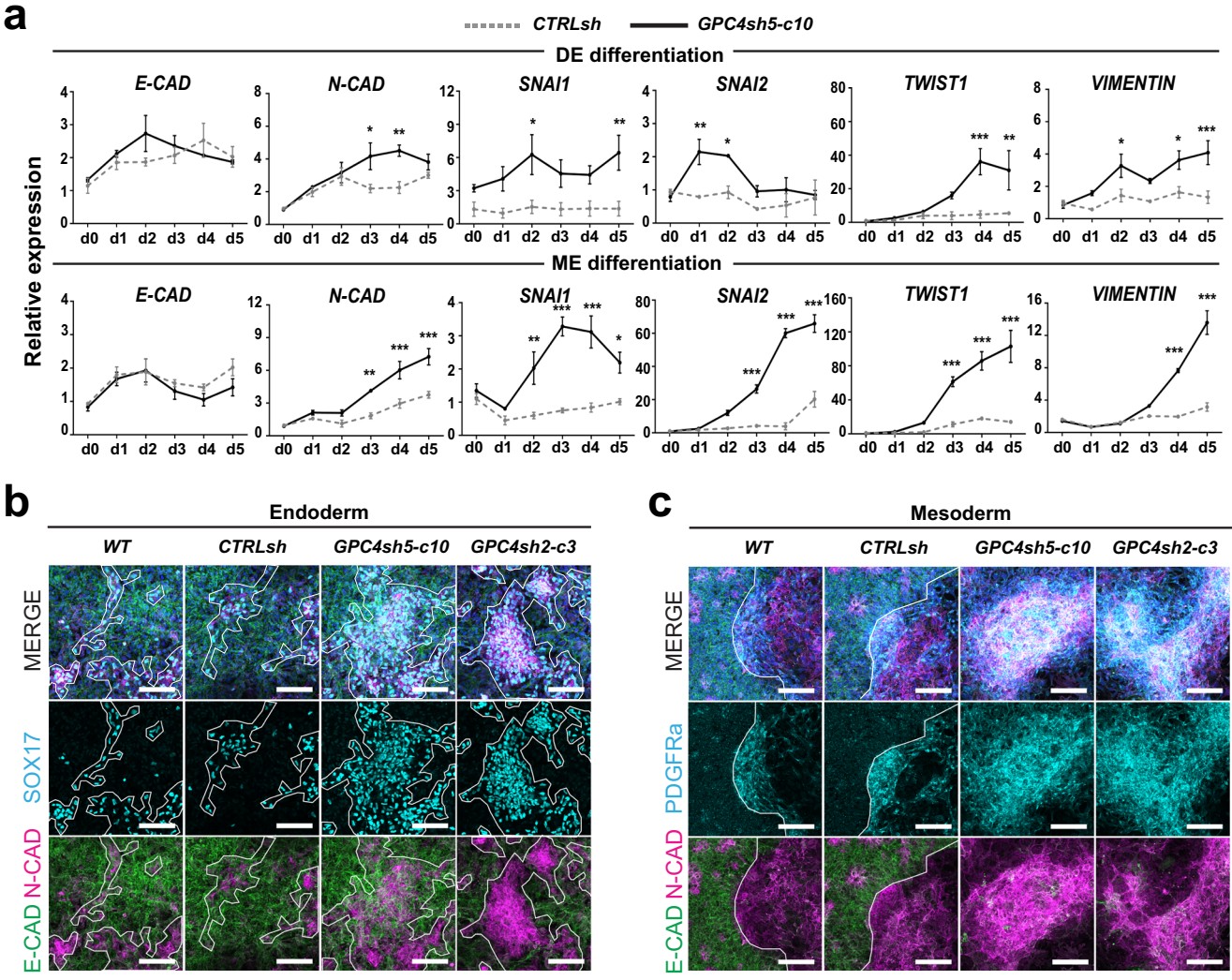

**Fig. 3 | Analysis of EMT in GPC4sh cells. a** RT-qPCR analysis of transcript levels of E-CAD, N-CAD, SNA1, SNAI2, TWIST1 and VIMENTIN in CTRLsh and GPC4sh5-c10 029 hiPSCs differentiated for 5 days into DE or ME. Top panel: DE differentiation and bottom panel: ME differentiation. Data were normalized to the mean of CTRLsh ΔCt at d0 and represented as mean ± SEM, *n* = 3. **b, c** Immunofluorescence analysis of N-CAD (magenta), E-CAD (green), SOX17 (cyan, DE) or PDGFRa (cyan ME) in WT,

CTRLsh, GPC4sh5-c10 and GPC4sh2-c3 029 hiPSCs differentiated for 3 days into DE (**b**) or ME (**c**), *n* = 2. Outlined areas highlight the DE and ME differentiated cells, scale bar: 100 µm. Statistical analysis for the overall figure: (**a**) two-way ANOVA followed by Sidak's multiple comparison test. *P*-values: (***) <0.001, (**) <0.01, (*) <0.05. For all panels "*n*" corresponds to the number of biological replicates unless stated otherwise. Source data are provided as a Source Data file.

Fig. 5c–e), thus excluding that proliferation and survival contribute to the increased differentiation efficiency in GPC4sh cultures.

### Loss-of-GPC4 impacts on self-organization of 2D gastruloids

To elucidate the differentiation properties of GPC4sh hiPSCs further, we took advantage of the 2D gastruloid system also known as micropattern that has emerged as a suitable in vitro system to study aspects of cell linage entry in hPSCs[7,8]. CTRLsh and GPC4sh hiPSCs were cultured as a monolayer on embryo-size circular adhesive micropatterns and exposed to BMP4. Both, CTRLsh and GPC4sh hiPSCs self-organized and gave rise to embryonic germ layers, arranged in concentric rings, in an ordered and reproducible manner. Interestingly, immunocytochemistry revealed differences in their differentiation profile. In particular, in comparison to CTRL hiPSCs, GPC4sh hiPSCs displayed an enlarged expression domain of BRACH cells expanding from the outer layer toward the center of the colony (Fig. 4a, b). This correlated with a narrower domain of SOX2 positive cells at the center of the micropattern in GPC4sh versus CTRL hiPSCs (Fig. 4a–d) as well as a reduced SOX2 intensity in these cells. Similarly, the SOX17 expression domain was spatially different, being more enlarged

towards the colony center at the expense of SOX2 (Fig. 4c, d). Thus, these observations on micropatterns are consistent with increased differentiation capability of GPC4sh hiPSCs towards distinct cell lineages such as MES (BRACH positive cells) and SOX17 derivatives. This would correlate with a cell fate switch of the presumptive SOX2 positive cells.

### Mesendoderm fate relies on epithelial integrity disruption

We next assessed whether disruption of the epithelial integrity in GPC4sh hiPSCs could underlie their increased differentiation potential into MES. This was addressed by restoring the epithelial integrity and polarity in GPC4sh hiPSCs, and by triggering disruption of the epithelial cell layer in CTRLsh cultures using specific chemical agents. Drug treated cells were then exposed to ACTIVIN A to instigate differentiation into the MES lineage. To restore the epithelial integrity of the GPC4sh cultures, we used lyso-phosphatidic acid (LPA), a small molecule that has been shown to cause an expansion of the apical domain of human neural progenitor cells[41]. Strikingly, a 24 h treatment of GPC4sh hiPSCs with LPA was sufficient to restore a TJ organization and an epithelial integrity similar to controls (Fig. 5a). Concomitant

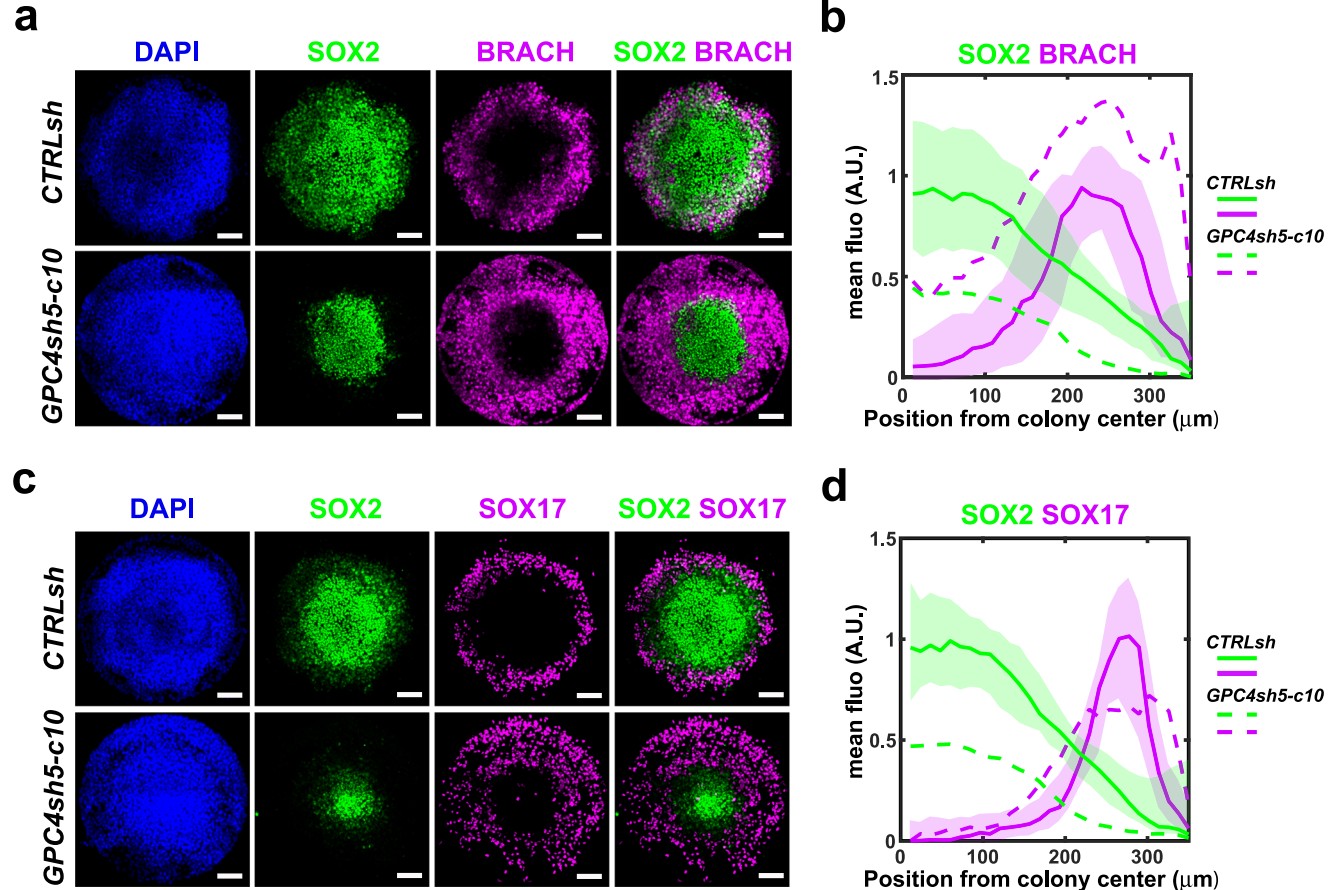

**Fig. 4 | GPC4 downregulation impacts lineage repartition in 2D gastruloids.**
**a** Representative 2D gastruloids immunostained for DAPI positive nuclei (blue), SOX2 (green) and BRACH (magenta) proteins 48h after the start of BMP4 stimulation (colony diameter 700 µm; scale bar 100 µm). **b** Quantification of SOX2 and BRACH radial profiles of expression in human 2D gastruloids of panel **a**. For each stain, radial profiles are normalized by the maximum of the control profile. Shaded areas represent the 1st and 3rd quartiles of the individual colony profiles with a line at the mean. CTRLsh: $n = 64$, GPC4sh $n = 67$ colonies from three biological replicates.

**c** Representative 2D gastruloids immunostained for DAPI positive nuclei (blue), SOX2 (green) and SOX17 (magenta) 48h after the start of BMP4 stimulation (colony diameter 700 µm, scale bar 100 µm). **d** Quantification of SOX2 and SOX17 radial profiles of expression in human 2D gastruloids of panel **c**. For each stain, radial profiles are normalized by the maximum of the control profile. Shaded areas represent the 1st and 3rd quartiles of the individual colony profiles with a line at the mean. CTRLsh: $n = 60$, GPC4sh $n = 62$ colonies from 3 biological replicates. Source data are provided as a Source Data file.

with the rescue of the TJ organization, LPA treatment also rescued the altered apical-basal polarity of GPC4sh hiPSCs characterizing the TJ disrupted areas (Fig. 5b), thus highlighting a crosstalk between TJ organization and apical-basal cell polarity. Conversely, when control hiPSCs were cultured for 24 h in the presence of Blebbistatin (Blebbi), a drug targeting Non-Muscle Myosin II[42,43], the epithelial integrity, assessed by analyzing TJs, was disrupted in a fashion similar to GPC4sh cultures (Fig. 5a, b). Furthermore, GPC4sh cultures in which the epithelial integrity was restored by LPA lose their enhanced MES differentiation properties and behave similar to control hiPSCs, as shown by the comparable expression levels of *MIXL1, GSC* and EOMES in both cell types (Fig. 5c, d). Instead, disruption of the epithelial integrity in control hiPSCs by Blebbi treatment enhanced their capability to undergo MES differentiation as in untreated GPC4sh hiPSCs, shown by the significant increase in *MIXL1, GSC* and EOMES expression (Fig. 5c, d). Collectively, these findings show that disruption of epithelial integrity enhances efficiency of differentiation into MES lineage.

Interestingly, by comparing the spatial distribution of cell patches with altered TJ organization with that of BRACH expressing cells in differentiating GPC4sh cultures at MES stage, we found a striking correlation into their spatial distributions. The correlation coefficient, calculated by quantifying the percentage of BRACH positive cells as a function of distance from epithelial disrupted areas, revealed a negative relationship between the two variables with slopes of -0.4082 and

-0.4325 for GPC4sh5 and GPC4sh2 lines, respectively (Fig. 5e, f). Taken together these results show that local disruption of epithelial integrity in the hiPSC layer regulates the onset of MES fate acquisition and its spatial pattern.

**Epithelial disruption enables onset of BMP/ACTIVIN signaling**
Decades of research on mammalian gastrulation have revealed that MES formation and spatial patterning is under the control of a signaling cascade initiated by BMP and involving activation of WNT and NODAL pathways[44–46]. Our results show that disruption of epithelial integrity in GPC4sh hiPSCs does not prime hiPSCs for MES fate acquisition. Yet, GPC4sh hiPSCs undergo efficient differentiation towards the MES lineage following ACTIVIN A stimulation. This raised the question of whether these structural changes in hiPSC epithelial layer may enhance their ability to activate BMP, ACTIVIN and/or WNT-β-CAT signaling cascade. In agreement with this possibility, prior studies using human embryonic SCs (hESCs) have shown that the epithelial organization and polarity of ESCs prevents accessibility of TGF-β superfamily ligands to their receptors located at the basolateral cell membrane as well as induction of the WNT-β-CAT signaling pathway[44–46]. As reported above, disruption of epithelial integrity in GPC4sh hiPSCs alters the apical-basal polarity of epithelial cells, thus enabling accessibility to proteins otherwise localized at the basolateral cell side (Fig. 1g and Supplementary Fig. 1h, i).

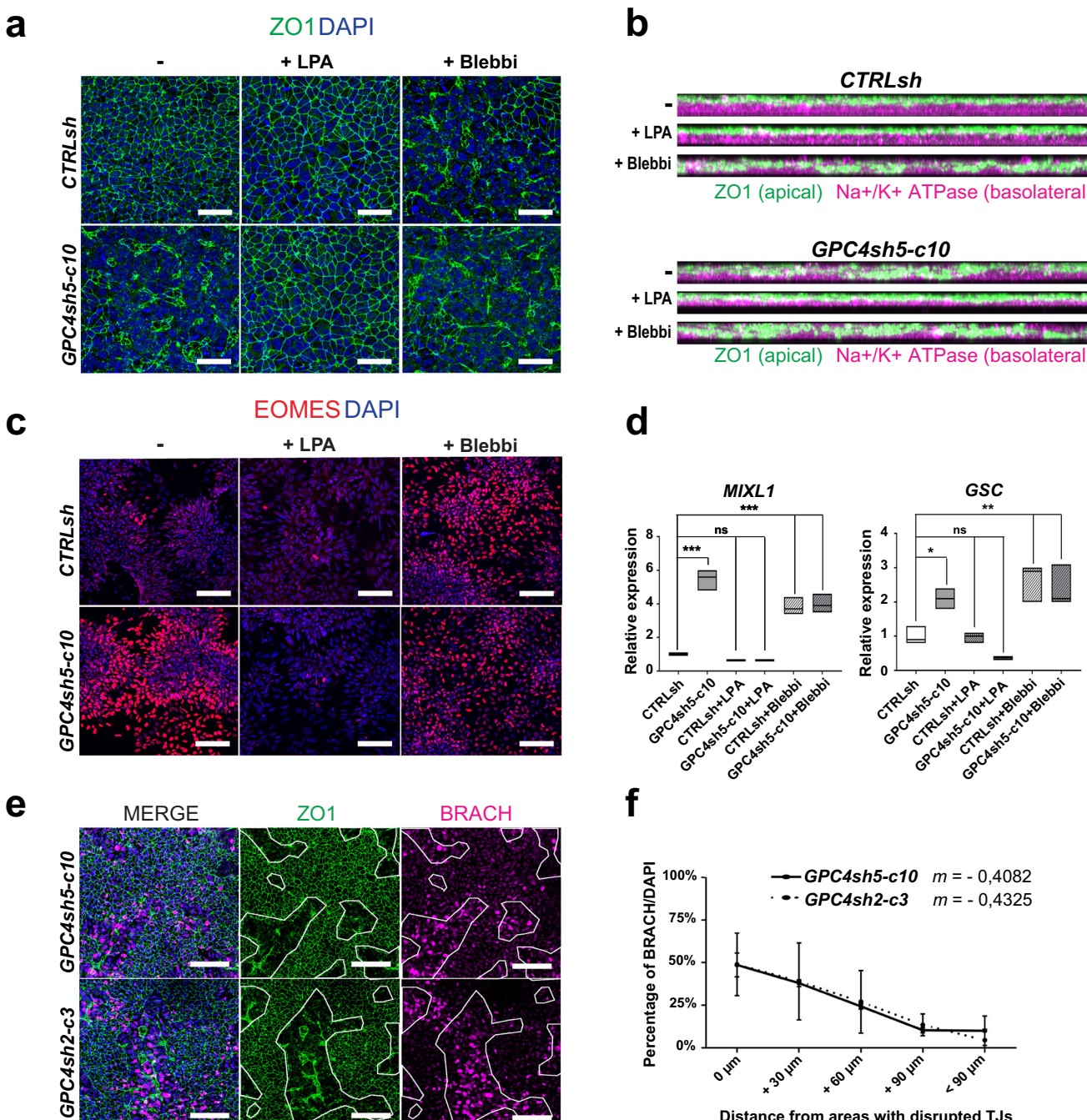

**Fig. 5 | MES fate acquisition relies on disruption of epithelial integrity.**
**a** CTRLsh and GPC4sh5-c10 029 hiPSCs were treated or not with LPA or Blebbi and subjected to immunofluorescence analysis of ZO1 (green), $n = 3$. Scale bar: 100 μm. **b** Immunofluorescence analysis of ZO1 (green) and Na$^+$/K$^+$ ATPase (magenta), in CTRLsh and GPC4sh5-c10 029 hiPSCs treated or not with LPA or Blebbi. Pictures are presented as lateral view of stained cells, $n = 3$. **c**, **d** CTRLsh and GPC4sh5-c10 029 hiPSCs treated or not with LPA or Blebbi were differentiated into MES and the expression levels of MES specific markers were analyzed by (**c**) immunofluorescence analysis of EOMES (red; $n = 3$; Scale bar: 100 μm.) and (**d**) RT-qPCR analyses of *MIXL1* and *GSC* transcript levels. Transcript levels were normalized to the ΔCt of reference WT control sample. Box plots represent the median with min and max values, $n = 3$. **e** GPC4sh5-c10 and GPC4sh2-c3 029 hiPSCs were differentiated into MES using

ACTIVIN A and subjected to immunofluorescence analysis of ZO1 (green) and BRACH (magenta). Outlined areas highlight the extent of zones with disrupted TJs, $n = 3$. Scale bar: 100 μm. **f** Percentage of BRACH positive cells as a function of distance from the zone with disrupted TJs were quantified from staining shown in **e**. 0 μm indicates cells located within the zones of disrupted TJs, +30 μm, +60 μm, +90 μm and >+90 μm indicate cells located at 0–30 μm, 30–60 μm, 60–90 μm or >90 μm away from zones of disrupted TJs, respectively. Slope of the curves are represented as *m*. Data are represented as mean ± SD. $n = 3$. Statistical analysis for the overall figure: (**d**) two-way ANOVA, followed by Dunnett's multiple comparison test, (**f**) linear regression analysis. *P*-values: (***) <0.001, (**) <0.01, (*) <0.05, ns not significant. For all panels "$n$" corresponds to the number of biological replicates unless stated otherwise. Source data are provided as a Source Data file.

To answer to this question, we analyzed the capacity of control and GPC4sh hiPSCs grown in self-renewal conditions to activate the BMP and ACTIVIN pathways upon a 1 h stimulation (immediate response). Cells responding to BMP4 and ACTIVIN A stimulations were visualized

by immunocytochemistry of nuclear phospho-SMAD1/5 (pSMAD1,5 S463/S465) and phospho-SMAD2 (pSMAD2 S465/S467), respectively[47]. After 1 h of stimulation, we found that the percentages of cells activating BMP and ACTIVIN signaling were ~3 and ~2-fold higher in

GPC4sh compared to control hiPSCs, respectively (Fig. 6a, b and Supplementary Fig. 6a, b). Additionally, image analysis revealed that most cells responding to BMP4 and ACTIVIN A stimulations localized in the areas with disrupted TJs (Fig. 6a and Supplementary Fig. 6a). This was confirmed by quantifying the percentage of pSMAD1,5 or pSMAD2 positive cells and their distance from epithelial disrupted areas. Strikingly, this quantification analysis revealed a strong negative correlation between these two variables, showing that they are interconnected (Fig. 6c and Supplementary Fig. 6c). Analysis of the second hiPSC line, AICS-0023, confirmed that most cells responding to BMP stimulation were localized in the areas with disrupted TJs in GPC4sh cultures compared to control cells (Supplementary Fig. 6d).

In line with these results, we found that when epithelial integrity of GPC4sh cultures was restored by LPA, the percentage of nuclear pSMAD1,5 positive cells was reduced to that of control hiPSCs (Fig. 6a, b). Conversely, when the epithelial integrity of control cells was disrupted by Blebbi treatment, the percentage of cells with activated BMP signaling was similar to that of GPC4sh5 hiPSCs and they mainly localized to areas with disrupted TJs (Fig. 6a, b). Finally, we observed a global activation of the BMP signaling in control and GPC4sh hiPSCs, when BMP4 was delivered from the basal side (Fig. 6d). These findings suggest that epithelial integrity in control hiPSCs prevents BMP4 signaling activation when ligands are provided from the apical side, likely due to inaccessible receptors. To examine this further, we analysed the localization of the BMPR1A and ACTR2B in CTRLsh and GPC4sh hiPSCs as these receptors are known to bind BMP4 and ACTIVIN A, respectively, and are the most expressed receptor types in our cells (from RNA seq analysis, see Data Availability). Immunocytochemistry revealed exposure of BMPR1A and ACTR2B receptors at the apical cell surface in areas with disrupted epithelial organization (Fig. 6e, f and Supplementary Fig. 6e, respectively). This was caused by changes in the ZO1 protein distribution (Fig. 6e, f and Supplementary Fig. 6e, respectively) and in the apical-basal cell polarity (Fig. 1g, Supplementary Fig. 1h, i). By performing molecular analysis of BMPR1A, ACTR2B expression, we did not find significant differences in transcripts and protein levels between CTRLsh and GPC4sh hiPSCs (Fig. 6g, h and Supplementary Fig. 6f, g). Similarly, the expression of BMP4, NODAL and CRIPTO (a NODAL co-receptor[35,48]) was unchanged in GPC4sh (Supplementary Fig. 6f, g). Taken together, these results show that local disruption of epithelial integrity does not impact on BMP and ACTIVIN receptor or ligand expression levels but enables perception of BMP4 and ACTIVIN A signaling proteins by cells located in these areas and activation of TGF-β signaling in a spatially regulated manner. This event will in turn trigger expression of MES markers in cells as well as their spatial pattern (Fig. 4 and Fig. 5e, f).

Considering that WNT-β-CAT signaling is an additional crucial regulator of MES differentiation, we assessed whether epithelial integrity influences the hiPSC ability to activate the WNT-β-CAT pathway upon stimulation. For these studies, we used immunocytochemistry to analyze the percentage of cells expressing LEF1, a co-factor of β-CAT that is a direct target of WNT signaling[49,50]. Control and GPC4sh hiPSCs were stimulated for 6 h with 50 ng/ml of WNT3A. By following LEF1 distribution, we observed between 50% and 60% of cells in both control and GPC4sh hiPSCs, respectively expressing LEF1, detected in a salt and pepper manner (Fig. 7a, b; WT 56.3% ± 6.4%; CTRLsh 57% ± 12.3%; GPC4sh5-c10 = 69% ± 9.5%; GPC4sh2-c3 = 73% ± 6.3%). Finally, there was no correlation between localization of LEF1 positive cells and patches of disrupted epithelial integrity in GPC4sh cultures ($m_{GPC4sh5} = -0.08776$ and $m_{GPC4sh2} = -0.05973$) (Fig. 7c). To analyze the WNT signaling response in more detail, we performed a dose-dependent experiment and stimulated cells with a WNT3A dosage ranging from 5 ng/ml (low activation) to 50 ng/ml (full activation). We did not observe significant differences in LEF1 expression and distribution at any concentration between control and GPC4sh hiPSCs (Fig. 7d–f). Moreover, the global nuclear intensity of LEF1 in cells did not change

between areas with intact or disrupted epithelial integrity (Fig. 7d–f). Thus, these results show that epithelial integrity does not impact on activation of WNT-β-CAT signaling nor its spatial localization, therefore highlighting its specific effect on BMP4 and ACTIVIN A pathways.

## Epithelial disruption prolongs BMP/ACTIVIN signaling

Recent studies using micropattern devices showed that the cellular response of PSC colonies to BMP stimulation varies over time. Indeed, in micropattern devices BMP4 responding cells become gradually restricted to the edge due to an increase in cell density and the acquisition of an epithelial cell morphology, marked by the presence of cell–cell contacts such as TJs. As a result, cells at the edge of the colony will express the BMP4 downstream effector pSMAD1,5 for a longer time than those in the colony center[11]. We therefore asked whether disruption of epithelial integrity influences maintenance of TGF-β signaling in hiPSC GPC4sh versus control cultures.

To address this issue, we performed a time course analysis to examine the ability of cells to perceive BMP4 and activated pSMAD1,5 in combination with the formation of TJs in control and shGPC4 hiPSCs after cell dissociation and seeding. Cells were dissociated and seeded at the density used for differentiation experiments ($1.1 \times 10^5$ cells/cm²) with ROCK inhibitor for 24 h. Then, ROCK inhibitor was removed and cells left to recover for 24 h prior to BMP and ACTIVIN stimulation. Following stimulation, cells were fixed at 0, 2, 6, 9, 12 and 18 h and immunostained with pSMAD1,5 antibodies. TJ organization was analyzed by ZO1 staining. After 6 h of stimulation, when hiPSC cultures have not yet recovered the TJ organization lost during the cell dissociation process, we observed a similar BMP4 response in both CTRLsh and GPC4sh5 hiPSCs with >85% of pSMAD1,5 positive cells (88.3%±3.6% in CTRLsh and 96%±2% in GPC4sh5 hiPSCs, Fig. 8a, b). However, as time progresses and cultures become denser, GPC4sh5 hiPSCs displayed qualitative and quantitative differences in comparison to CTRLsh cultures. Whereas the percentage of cells with activated BMP signaling pathway decreased rapidly in CTRLsh cultures, this reduction occurred to a lesser degree in GPC4sh hiPSCs leading to greater number of pSMAD1,5 positive cells in GPC4sh versus control cultures between 9–18 h (9 h: 66.7% ± 4.9% in CTRLsh and 95% ± 2.3% in GPC4sh hiPSCs; 12 h: 22.3% ± 1.2% in CTRLsh and 51.7% ± 7.5% in GPC4sh hiPSCs; 18 h: 7.7% ± 1.2% in CTRLsh and 15.7% ± 4.3% in GPC4sh hiPSCs Fig. 8a, b). Similar effects were observed when CTRLsh and GPC4sh5 hiPSCs were stimulated with ACTIVIN A (Supplementary Fig. 7a–c). Interestingly, analysis of ZO1 staining overtime revealed striking changes in the kinetics of TJ formation between CTRLsh and GPC4sh hiPSCs. In particular, GPC4sh hiPSCs displayed a delay in generating TJs as revealed by the lower percentage of ZO1 organized areas in GPC4sh versus CTRLsh at 9 and 12 h post seeding (9 h: 83.5% ± 4.6% in CTRLsh and 34.2%±6% in GPC4sh cultures; 12 h: 91.2%±2.7% in CTRLsh and 68% ± 8,4% in GPC4sh cultures; Fig. 8a, b). Results also highlighted that most GPC4sh cells with sustained activation of BMP4 signaling localize to the areas presenting disrupted ZO1 pattern. Therefore, we concluded that a delay in TJ formation in GPC4sh cultures accounts for the cellular/tissue mechanism that promotes a prolonged/sustained activation of BMP4 and ACTIVIN A signaling pathways in these cells.

To deepen our analysis, we performed stimulation experiments in the presence of Blebbi to perturb TJ organization and LPA to promote TJ formation. Blebbi or LPA were added to the cultures at a time point in which CTRLsh and GPC4sh5 hiPSCs activate the BMP4 signaling pathway to the same extent (6 h; Fig. 8a). Analysis of pSMAD1,5 positive cells at 12 and 18 h revealed that Blebbi treatment triggered TJ disorganization and activation of the BMP4 signaling pathway within disrupted areas (Fig. 9a, b and Supplementary Fig. 7d). In contrast, LPA significantly reduced the presence of TJ disrupted areas and blocked activation of the BMP4 signaling pathways in both CTRLsh and GPC4sh hiPSCs (Fig. 9a, b and Supplementary Fig. 7d). As addition of LPA rescues the shutdown of the BMP4 signaling pathways in GPC4sh

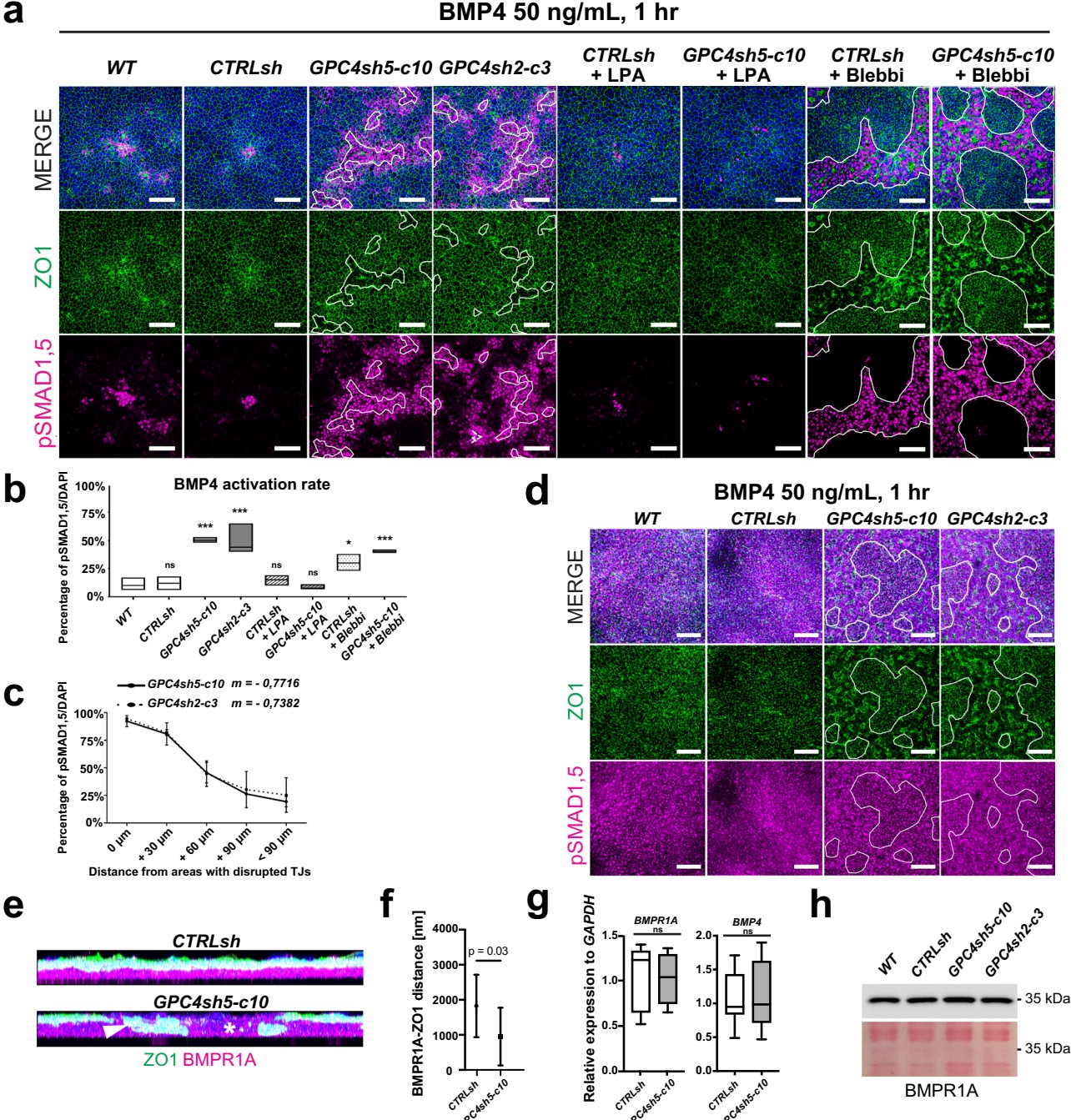

**Fig. 6 | Disruption of epithelial integrity enables activation of TGF-β signaling.**
**a** Cultures of WT, CTRLsh, GPC4sh5-c10 and GPC4sh2-c3 hiPSCs treated or not with LPA or Blebbi were stimulated for 1 h with BMP4 (50 ng/mL), and analyzed for pSMAD1,5 (magenta) and ZO1 (green) proteins by immunocytochemistry. Dashed areas highlight zones with disrupted TJs, n = 3. Scale bar: 100 μm. **b** Percentages of pSMAD1,5 positive cells quantified from staining in **a**. Box plots represent the median with min and max values, n = 3. **c** Percentages of pSMAD1,5 positive cells as a function of distance from zones with disrupted TJs quantified from staining in **a**. 0 μm indicates cells located within zones with disrupted TJs, +30 μm, +60 μm, +90 μm and <+90 μm indicate cells located at 0 to 30 μm, 30–60 μm, 60–90 μm or >90 μm away from zones with disrupted TJs, respectively. Data are represented as mean ± SD, n = 3. **d** Immunofluorescence analysis of pSMAD1,5 (magenta) and ZO1 (green) in the indicated hiPSC lines grown in transwell plates and stimulated from the basal side with BMP4 (50 ng/mL). Outlined areas highlight zones of disrupted

TJs, n = 1. Scale bar: 100 μm. **e** Immunofluorescence analysis of ZO1 (green) and BMPR1A (magenta), in CTRLsh and GPC4sh5-c10 hiPSCs. Pictures are presented as lateral view of stained cells, n = 3. **f** Quantification of BMPR1A receptor distance to the plane of ZO1 contacts in CTRLsh and GPC4sh5-c10 hiPSCs. Data are represented as mean +/− SD of n = 10 frames of view. **g** RT-qPCR analysis of *BMPR1A* and *BMP4* transcripts in CTRLsh and GPC4sh5-c10 hiPSCs. Data are represented as box and whisker plot with a line at the median, n = 8. **h** Protein extract of the indicated hiPSC lines were analyzed by Western-blot using BMPR1A antibodies. Membrane ponceau staining was used as loading control, n = 2. Statistical analysis for the overall figure: (**b**) two-way ANOVA, followed by Dunnett's multiple comparison test, (**f**) unpaired *t*-test. *P*-values: (***) <0.001, (**) <0.01, (*) <0.05, ns not significant. For all panels "*n*" corresponds to the number of biological replicates. Source data are provided as a Source Data file.

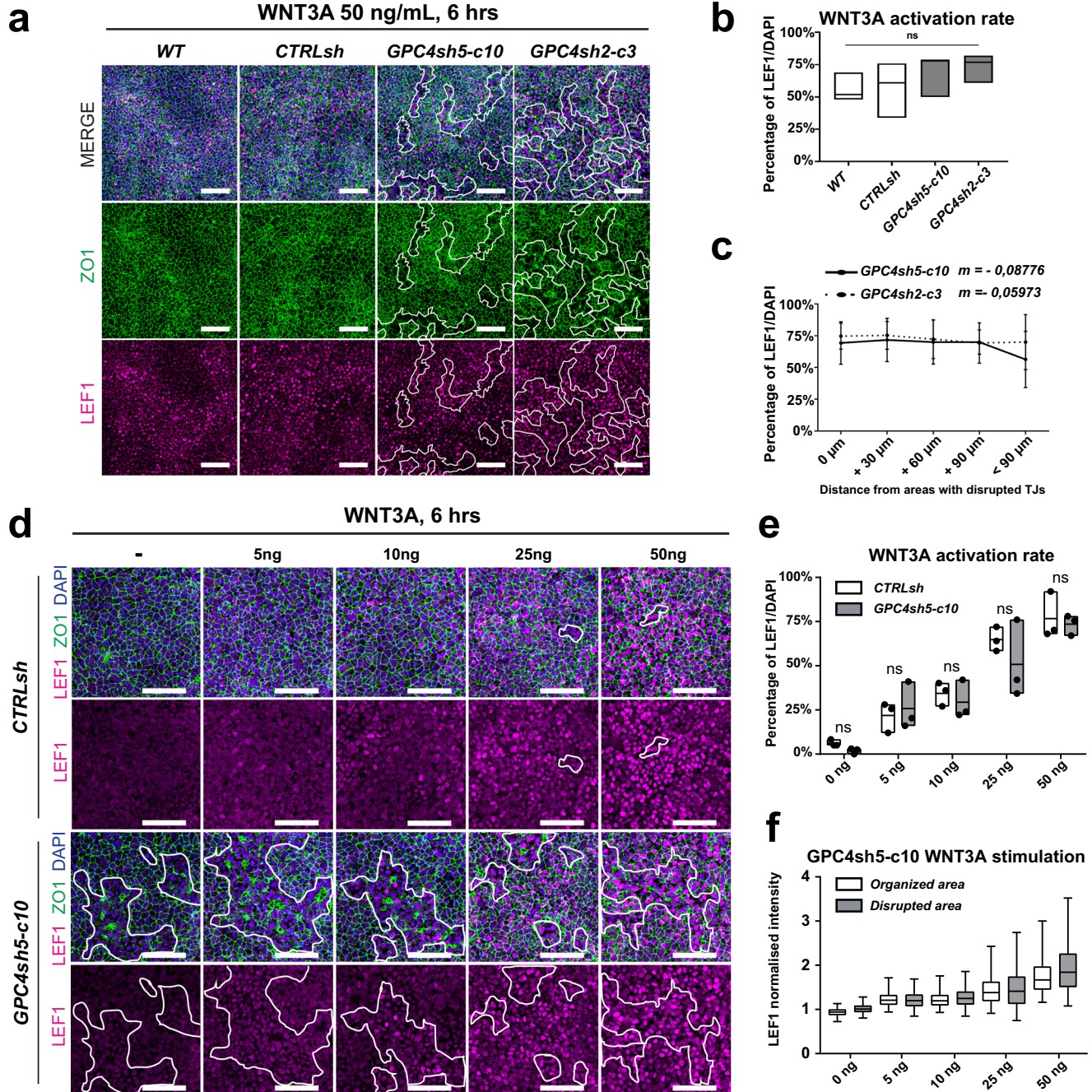

**Fig. 7 | Disruption of epithelial integrity does not perturb WNT signaling activation. a** Cultures of WT, CTRLsh, GPC4sh5-c10 and GPC4sh2-c3 029 hiPSCs were stimulated for 6 h with 50 ng/mL of WNT3A, and analyzed for LEF1 (magenta) and ZO1 (green) proteins. Dashed areas highlight the disturbed TJs, *n* = 3. Scale bar: 100 μm. **b** Percentages of LEF1 expressing cells were quantified from staining in **a**. Box plots represent the median with min and max values, *n* = 3. **c** Percentages of LEF1 expressing cells as a function of the distance from zones with disrupted TJs quantified in GPC4sh5-c10 and GPC4sh2-c3 029 hiPSCs from staining shown in **a**. 0 μm indicates cells located within the zones of TJs disruption, +30 μm, +60 μm, +90 μm and <+90 μm indicate cells located at 0–30 μm, 30–60 μm, 60–90 μm or >90 μm from the zone of TJs disruption respectively. Data are represented as mean ± SD, *n* = 3. **d** Cultures of CTRLsh and GPC4sh5-c10 029 hiPSCs were stimu- lated for 6 h with 5, 10, 25 and 50 ng/mL of WNT3A, and then analyzed for LEF1

(magenta) and ZO1 (green) proteins. Outlined areas highlight the disturbed TJs, scale bar: 100 μm, *n* = 3. **e** Percentages of LEF1 expressing cells in CTRLsh and GPC4sh5-c10 029 hiPSCs from staining shown in **d**. Box plots represent the mean with min and max values, *n* = 3. **f** Analysis of LEF1 normalized intensity in organized and disrupted areas of GPC4sh5-c10 029 hiPSCs from staining shown in **d**, represented as a box and whisker plot; whiskers represent 1st and 99th per- centiles, *n* = 3 with 7957 to 26429 single nucleus values per condition. Statistical analysis for the overall figure: (**b**) two-way ANOVA followed by Dunnett's multiple comparison test, (**c**) linear regression analysis, (**e**) two-way ANOVA followed by Sidak's multiple comparison test. *P*-values: (***) <0.001, (**) <0.01, (*) <0.05, ns not significant. For all panels "*n*" corresponds to the number of biological replicates unless stated otherwise. Source data are provided as a Source Data file.

cultures, we can conclude that the underlying mechanism of intracel- lular signaling prolongation/extension in GPC4sh hiPSCs is mainly due to their TJ organization properties upstream of any intracellular sig- naling event. Of note, similar results were observed when CTRLsh and

GPC4sh5 hiPSCs were stimulated with ACTIVIN A in the presence of LPA and Blebbi (Supplementary Fig. 7a–c).

Taken together, these results highlight a key role for GPC4 in modulating the morphological organization of the hiPSC epithelial

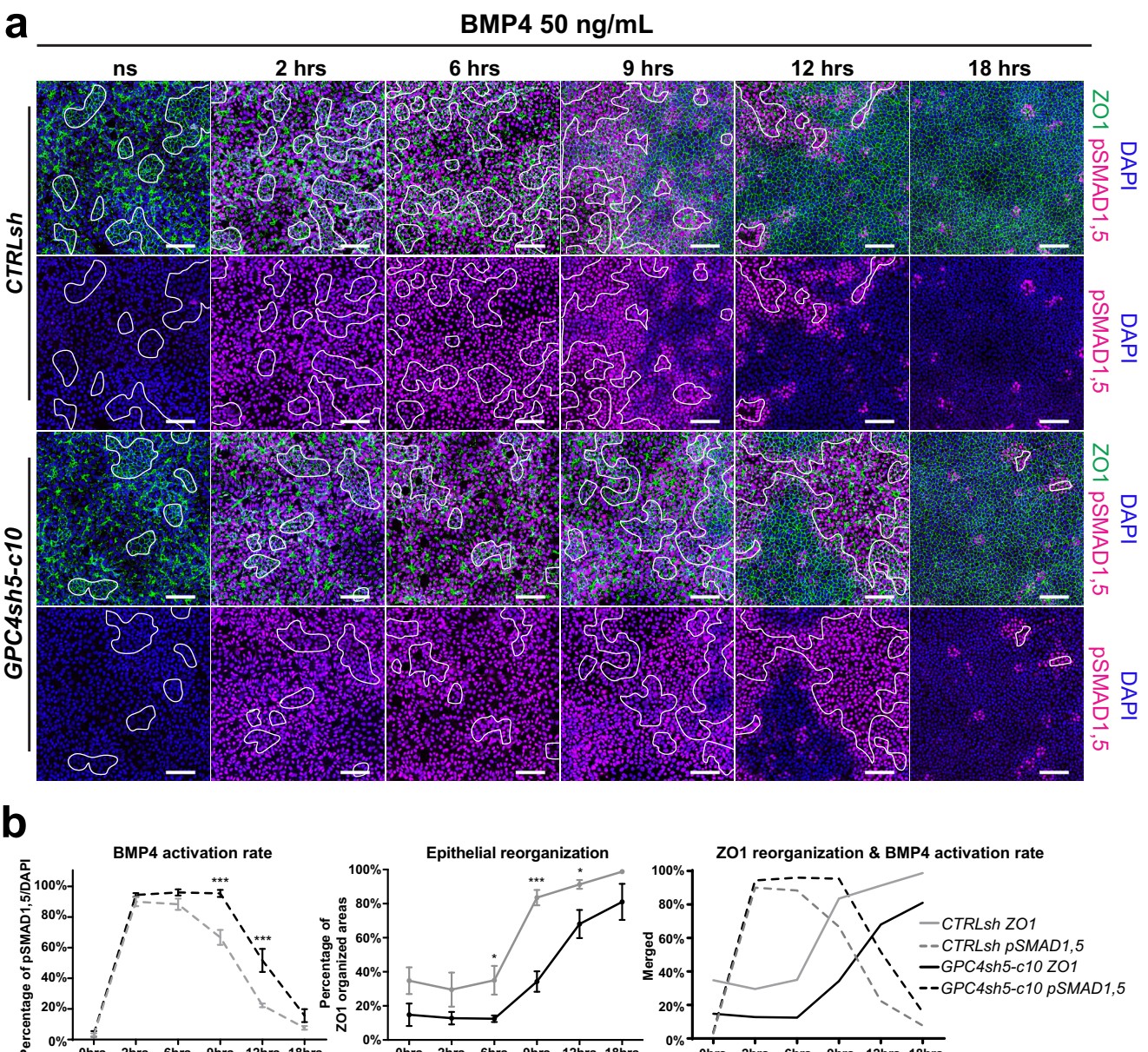

**Fig. 8 | GPC4 modulates hiPSC epithelial layer organization by impacting on the kinetics of TJ formation. a** Time course analysis of TJ formation in CTRLsh and GPC4sh hiPSCs after cell dissociation and seeding in combination with the analysis of pSMAD1,5 expression in response to BMP4 stimulation for 2, 6, 9, 12 and 18 h. Cells were subjected to immunofluorescence analysis of pSMAD1,5 (magenta) and ZO1 (green) proteins, $n = 3$. Scale bar: 100 µm. **b** Quantitative analysis from the staining shown in **a** of pSMAD1,5 positive cells (left panel) and of ZO1 organized areas (middle panel) in CTRLsh and in shGPC4 hiPSC cultures. Note the different kinetics in GPC4sh cultures versus controls. The right panel show the combination of the left and middle panels to highlight the correlation between the percentage of pSMAD1,5 positive cells and of ZO1 organized area. Data are represented as mean ± SEM, $n = 3$ for pSMAD1,5 staining and $n = 4$ for ZO1 staining. Statistical analysis: Statistical analysis: (**b**) two-way ANOVA followed by Sidak's multiple comparison test were used to analyzed pSMAD1,5 counts and unpaired *t*-test were used to analyzed ZO1 organized areas *P*-values: (\*\*\*) <0.001, (\*) <0.05. For all panels "*n*" corresponds to the number of biological replicates unless stated otherwise. Source data are provided as a Source Data file.

layer by impacting on the kinetics of TJ formation. Moreover, they show that a protracted disruption of the hiPSC epithelia layer maintains activation of BMP4 and ACTIVIN A signaling in the cells located in disrupted areas for a longer time frame, which might impact on specificity and robustness of cell fate acquisition. These outcomes also provide additional mechanistic insights on how MES triggering signals could be temporally controlled.

## Discussion
Unravelling the processes of primitive streak formation and epiblast patterning in mammals remains a great challenge, especially for

human embryos. In vitro models of the human epiblast based on hPSCs are offering the unique opportunity to decompose these complex processes and allow independent manipulation of the underlying mechanisms. Current consensus focuses on the concept that these early embryonic processes are controlled by feedback interactions between biochemical signals (morphogens and their inhibitors/activators) and tissue organization (epithelial cell polarity, adhesion and cytoskeleton tension). In the present study we explored how these instructive cues functionally interact and whether they operate in a hierarchical manner. Here we report that hiPSCs with reduced protein levels of the morphogen regulator GPC4 represent a relevant cellular

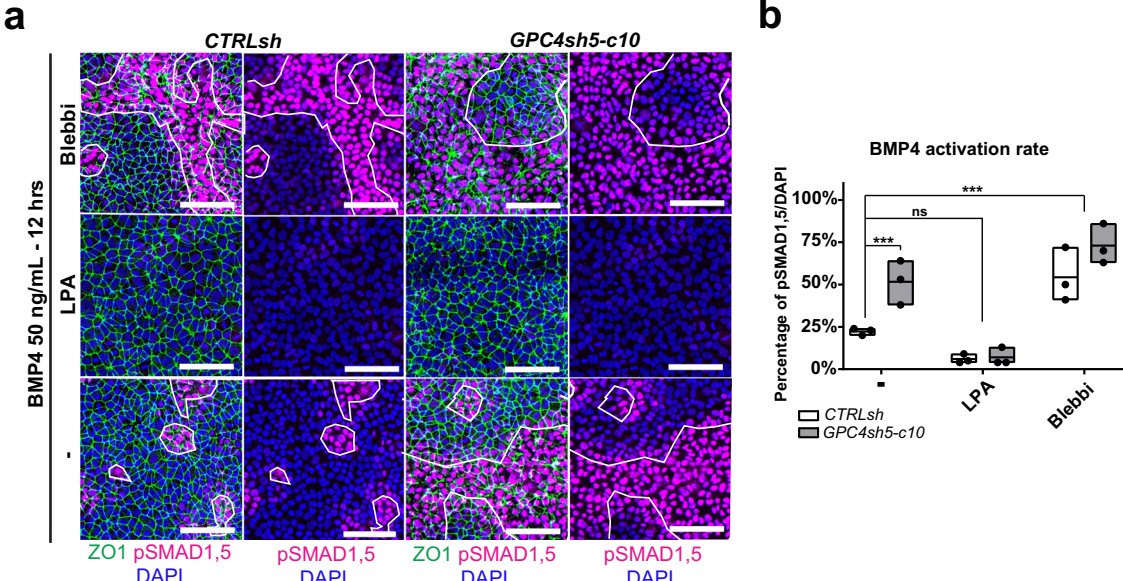

**Fig. 9 | Disruption of epithelial integrity prolongs TGF-β signaling activation.** **a** Analysis of pSMAD1,5 and ZO1 expression in control and shGPC4 cells stimulated for 12 h with 50 ng/mL of BMP4 in the presence of Blebbistatin (Blebbi) and LPA (added 6 h after stimulation onset) to impair or promote TJ formation. Cultures were subjected to immunofluorescence analysis of pSMAD1,5 (magenta) and ZO1 (green) proteins, n = 3. Outlined areas in the overall panels highlight the extent of zones of disrupted TJs, scale bar: 100 μm. **b** Percentages of pSMAD1,5 expressing cells were quantified from staining shown in **a**. Box plots represent the mean with min and max values, n = 3. Statistical analysis: (**b**) two-way ANOVA followed by Sidak's multiple comparison test. P-values: (***) <0.001, ns not significant. For all panels "n" corresponds to the number of biological replicates unless stated otherwise. Source data are provided as a Source Data file.

system to investigate the physical effects of disrupted epithelial cell polarity and architecture on the cellular response to TGF-β proteins at the human MES onset. We demonstrated that GPC4 downregulation in hiPSCs results in a distinct morphological organization of hiPSC monolayers wherein alteration of TJs leads to a mosaic hiPSC structural pattern with areas of disrupted epithelial integrity. This phenotype alters the apical-basal cell polarity causing exposure of proteins in the basolateral cell compartment. Moreover, it influences the spatial regulation of cell fate acquisition, as cells within areas of disrupted epithelial integrity become sensitive to BMP4 and ACTIVIN A protein stimulation and acquire a MES fate (e.g. BRACH and EOMES expression). This is accompanied by changes in the temporal dynamics of cell signaling responses to BMP4 and ACTIVIN A given the protracted kinetics of TJ formation in GPC4sh hiPSCs. Thus, our results demonstrate how distinct changes in epiblast cell polarity and tissue integrity can foster the spatial and temporal response of epiblast cells to the MES-initiating signal and the subsequent differentiation onset in a precisely controlled manner.

Our experiments involving apical and basal stimulation of CTRLsh and GPC4sh hiPSCs with BMP4 and ACTIVIN A proteins provide insights into the mechanisms underlying spatial control of these signals. Under culture conditions designed for pluripotency maintenance, BMP4 and ACTIVIN A reception in CTRLsh hiPSCs is restricted to the basolateral cell domain, as cells activate BMP4 and ACTIVIN A signaling only following basal ligand stimulation. This is consistent with recent reports in which the use of a 2D micropatterned hESC-based epiblast model revealed that hESCs localize their BMP receptors to the basolateral domain just below the tight junctions, making them non-responsive to apically applied BMP4[11]. In contrast, the disrupted epithelial integrity of GPC4sh hiPSCs leads to an impaired apical-basal cell polarity in TJ disrupted areas with concomitant apical exposure of BMPR1A and ACTR2B. This enables local perception of BMP4/ACTIVIN A and signaling pathway activation independently of whether the ligand is supplied from the basal or apical cell side. Thus, the morphological organization of the GPC4sh hiPSC layer with disrupted TJs enables increased accessibility of ligands to receptors. This enhanced

capability of GPC4sh hiPSCs to perceive BMP4 signal is consistent with the increased differentiation observed on micropatterned substrates. Interestingly, the expansion of the BRACH and SOX17 spatial domains in differentiating GPC4sh cells might indicate a potential in vivo bias in embryos lacking GPC4 towards the mesendoderm lineage. Thus, GPC4 and epithelial organization might participate to regulate the ectoderm/mesendoderm ratio. 3D gastruloids generated from hPSCs have emerged as an in vitro model system of the anteroposterior organization of the early human body plan. It will be interesting to examine whether disruption of the epithelial organization in GPC4sh hiPSCs affects elongation of gastruloids along an anteroposterior axis in addition to impacting cell lineage fate.

A similar pattern of basolateral BMP4 receptor localization below TJs has also been reported in vivo in the developing mouse epiblast[6]. Similar to the finding we describe here, lack of TJs at the border between the extraembryonic ectoderm and posterior epiblast enables BMP ligands secreted into the proamniotic cavity to access receptors at the posterior epiblast edge, and to trigger PS formation at this level[6]. Our results together with these embryological studies point to a feedback loop between epiblast morphology and signaling, in which epiblast epithelial integrity controls spatial activation of TGF-β family members' signaling and patterning. In this context, GPC4sh hiPSCs may represent an optimal human cellular system to analyze further this feedback interaction and to experimentally dissect mechanisms controlling epiblast cell polarity and tissue morphology.

Whereas GPC4 is not required for TJ organization, our results highlight a role for GPC4 in regulating the kinetics of TJ formation as revealed by time course experiments showing a delay of TJ assembly in GPC4sh versus control hiPSCs after cell dissociation and seeding. As TJ restricts BMP and ACTIVIN signaling pathways over time, it is tempting to speculate that fluctuation of GPC4 protein levels might change the kinetics of TJ formation in mammalian epiblast, which could in turn impact on the temporal dynamics of instructive signaling pathways. Thus, these overall findings provide additional mechanistic insights on how MES triggering signals could be temporally controlled and on how

tissue organization might impact on specificity and robustness of cell fate acquisition.

In contrast to TGF-β proteins, disruption of epithelial integrity does not regulate spatial activation of WNT-β-CAT signaling in our experimental conditions, as shown by a rather homogeneous activation of LEF1 within the GPC4sh hiPSC layer following WNT3A stimulation. These results are consistent with previous works showing that WNT receptors are accessible to WNT ligands from both the apical and the basolateral cell side[46]. Nevertheless, activation of WNT-βcat signaling is sensitive to changes in tissue-level forces, which are dependent on tissue organization. As modification of TJ distribution affects global tissue tension, we cannot exclude that our experimental conditions prevented detection of altered hiPSC response to WNT3A stimulation associated with epithelial integrity. Our studies were conducted on cells cultured on stiff substrates (glass/plastic) on which possible changes in tissue tension induced by loss of TJs might be buffered. Studies by others have shown that in hESCs cultured on soft substrates, activation of WNT-βCAT signaling correlates with high tissue-level forces[13,51]. Thus, it could be relevant to perform future studies culturing GPC4sh hiPSCs on soft substrates, such as hydrogels, to assess whether disruption of epithelial integrity modulates WNT response by modifying global tissue tension.

In contrast to hiPSCs, lack of GPC4 in mouse ESCs impairs activation of the WNT signaling pathway[15]. This apparent discrepancy might arise from the different requirements of WNT signaling in mouse ESCs versus hiPSCs. In support of this possibility, mouse ESCs and hPSCs show fundamental differences in colony shape, growth rate, surface markers, and developmental potential in addition to global molecular signatures and signaling pathways governing self-renewal and differentiation[52]. Thus, our study might highlight distinct molecular requirements for mouse ESC and hPSC biological processes.

Interestingly, disruption of epithelial integrity in the GPC4sh hiPSC layer does not perturb self-renewal and pluripotency neither promotes premature expression of EMT markers or lineage specific genes (present studies and in[14]). Thus, alterations in TJs, although priming cells towards morphogen perception, are not sufficient to prime differentiation or lineage fate choices. It is tempting to speculate that this epithelial disruption is not sufficient to achieve a threshold level required for differentiation. Alternatively, unknown factors could buffer such drastic alteration in epithelial organization. The observed maintenance of stemness/pluripotency in GPC4sh hiPSCs is consistent with recent studies on hESCs carrying a genetic knockout of the human cell–cell adhesion protein, E-CAD. Nevertheless, these cells show a transient activation of lineage-specific genes and preferential differentiation towards MES, thus suggesting that loss of E-CAD impacts on lineage fate decisions[32,53].

Our results also disclose an additional role of GPC4 in maintaining epithelial cell tissue integrity. Previous studies performed on Zebrafish and Xenopus embryos have highlighted GPC4 function during ME convergence extension movements at gastrulation in which GPC4 controls polarity and directed migration of mesodermal cells[18,20,54,55]. Moreover, in Zebrafish embryos GPC4 promotes DE cell polarity by influencing localization of N-CAD on the cell surface through regulation of Rab5c-mediated endocytosis[56]. Future experimental settings will clarify whether mechanisms involving GPC4-mediated endocytosis of membrane proteins participating to cell–cell and cell-matrix contacts underlie the epithelial integrity defects found in GPC4sh hiPSCs.

In conclusion, our results highlight an additional mechanism by which tissue architecture can restrict morphogen sensing and position cell fate changes within space. Our differentiation analysis of GPC4sh hiPSCs reveals their increased differentiation propensity in ME and DE compared to conventional hiPSCs. These finding support the possibility that targeting GPC4 in different hiPSCs may provide an additional strategy to overcome the technical difficulties involved in their

differentiation, thus providing a tool to promote hiPSC application for disease modelling, drug screening and regenerative medicine. Furthermore, GPC4sh hiPSCs may provide a platform where differentiation relevant processes can be followed and deconstructed. GPC4sh hiPSCs are thus a means to obtain new insights into developmental processes and for advancing into regenerative medicine. We expect that this mechanism of modulating epithelial integrity by altering TJs will be of more general relevance in other physiological and pathological events in which there is a high degree of epithelial cell plasticity, such as remodelling of the epithelial barriers, tissue inflammation and tumorigenesis.

## Methods

### Human induced pluripotent stem cell lines and culture

WT 029 hiPSCs were a courtesy of Michael Kyba (University of Minnesota)[57]. The AICS-0023 previously used were from Coriell Institute for Medical Research. The generation of GFPsh hiPSCs and GPC4sh hiPSCs was previously described[14]. hiPSCs were maintained on Matrigel-coated plates (Corning, BV 354277) within mTeSR1 medium (mTeSR™1 (Stemcell Technologies 85850)), 1% Penicillin/Streptomycin (Invitrogen, ref. 15140122). In order to maintain hiPSC cultures not exceeding 80% of confluency, cells were passaged at 1/8 ratio every 3 or 4 days with Accumax (Millipore, ref. SCR006).

### Alkaline phosphatase assay

Alkaline Phosphatase (AP) detection has been performed with the Alkaline Phosphatase Staining Kit II (Stemgent, ref. 00–0055). Briefly, hiPSCs were seeded on Matrigel-coated coverslips until reaching a maximum of 80% of confluency. Cells were incubated with Fix Solution at room temperature for 5 min and then washed with PBS. Afterwards, cells were incubated with the AP substrate solution in the dark, at room temperature for 10 min. The reaction was stopped by aspirating the AP Substrate solution and washing the wells twice with PBS. Images were captured on a Zeiss Stereo microscope.

### Cell cycle analysis

hiPSCs were seeded on Matrigel-coated plates and analyzed at 30–50% of cells confluency. Nuclei were extracted with lysis buffer (Chemometec) and incubated with 10 μg/ml of DAPI for 5 min at 37 °C. Nuclear DAPI intensity was measured with the Cytometer NucleoCounter® NC-3000™ (Chemometec) and cell cycle was analyzed with the Nucleo-View NC-3000™ program (Chemometec).

### Differentiation assays into MES, DE and ME lineages

Differentiation experiments of hiPSCs into DE and ME lineages were performed by following the protocols published by[33,34] respectively. To generate MES, DE and ME lineages, hiPSCs were seeded on Matrigel-coated plates or coverslips at a density of $1.1 \times 10^5$ cells/cm² in mTeSR1 medium supplemented with 10 μM of ROCK inhibitor Y-27632 (Ri; Tocris, ref. 1254). To induce MES differentiation, hiPSCs were washed twice with RPMI (Invitrogen, ref. 21875034) to remove self-renewal growth factors and treated for 1 day with ACTIVIN A (R&D, ref. 338-AC, 100 ng/mL) in DE medium (RPMI 1640, 1% Penicillin/Streptomycin, 1% L-Glutamine (Gibco, ref. 25–030)). DE differentiation was induced by treating MES differentiated cultures for 2 days with ACTIVIN A (R&D, ref. 338-AC, 100 ng/mL) in DE medium supplemented with 0.2% of FBS (Hyclone, ref. SH30071.02E). ME differentiation was induced by treating PS differentiated cultures for 2 days with BMP4 (R&D, ref. 314-BP, 10 ng/mL) in ME differentiation medium (RPMI 1640, 2% B27 without Insulin (Gibco, ref. A1895601)), 1% Penicillin/Streptomycin, 1% L-Glutamine (Invitrogen, ref. 25030149)). Medium was replaced every day. The protocol reporting medium replacement is summarized in Supplementary Table 3.

For experiments involving drug treatments, hiPSCs were seeded on Matrigel-coated plates or coverslips at a density of

$0.25 \times 10^5$ cells/cm² within mTeSR1 medium supplemented with Ri. The next day, medium was replaced by fresh mTeSR1 medium. The day after rescue of epithelial integrity was induced by treating hiPSCs for 1 day with LPA (Stemcell Technologies, ref. 72694, 5 μM). Chemical disruption of epithelial integrity was induced by treating hiPSCs for 1 day with Blebbistatin (Sigma, ref. B0560, 10 μM). MES differentiation was subsequently induced by washing cultures twice with RPMI and treating hiPSCs for 1 day with ACTIVIN A (100 ng/mL) in DE medium supplemented with LPA (5 μM) or Blebbistatin (10 μM). Medium was replaced every day. The protocol reporting medium replacement is summarized in Supplementary Table 4.

### 2D micropatterns
Cells were seeded at a density of 600 cell/mm² on substrates micro-patterned with laminin 521 using the microcontact printing techniques[58,59] (see source files for full methods). BMP4 stimulation was started 15–24 h after seeding. Medium was refreshed every day. Colonies were fixed after 48 h and analysed by immunocytochemistry.

### Stimulation assays
Stimulation assays were performed on confluent hiPSC cultures maintained in mTeSR1 medium. HiPSCs were seeded at $1.5 \times 10^5$ cells/cm² on Matrigel-coated coverslips in mTeSR1 medium supplemented with Ri. The next day, medium was replaced by fresh mTeSR1 medium. The day after hiPSCs cultures were stimulated for 1 h with BMP4 (R&D, ref. 314-BP, 50 ng/mL) or ACTIVIN A (100 ng/mL), and 6 h with WNT3A (R&D, ref. 5036-WNT, ranging from 5 ng/mL to 50 ng/mL) in mTeSR1 medium. Cells were then fixed and signaling pathways activation was assessed through immunocytochemical analyses. Medium replacement protocols are summarized in Supplementary Table 5.

For experiments of rescue and chemical induction of epithelial integrity, hiPSCs were seeded on Matrigel-coated coverslips at a density of $0.75 \times 10^5$ cells/cm² in mTeSR1 medium supplemented with Ri. The next day, medium was replaced by fresh mTeSR1 medium. The day after, medium was replaced for 1 day by mTeSR1 medium supplemented with 5 μM of LPA or 10 μM of Blebbistatin. Finally, confluent cultures were stimulated for 1 h with BMP4 (50 ng/mL) in mTeSR1 medium, and then fixed and analyzed by immunocytochemistry.

For long-term stimulation experiments of BMP and ACTIVIN pathways, hiPSCs were seeded on Matrigel-coated coverslips at a density of $1.1 \times 10^5$ cells/cm² in mTeSR1 medium supplemented with Ri. The next day, medium was replaced by fresh mTeSR1 medium. The day after, hiPSCs cultures were stimulated with BMP4 (50 ng/mL) or ACTIVIN A (100 ng/mL) in mTeSR1 medium. hiPSCs cultures were then fixed and analyzed by immunocytochemistry at 0, 2, 6, 9, 12, and 18 h following stimulation.

### RNA isolation, cDNA synthesis and quantitative PCR
Total RNA was isolated with RNeasy Min Elute clean-up kit (Qiagen, ref. 74204), according to the manufacturer's protocol. Concentration and RNA quality were evaluated by NanoDrop (ND-1000 Spectro-photometer; Thermofisher Scientific). 600 ng of RNA was reverse-transcribed (RT) to generate cDNA using iScript reverse transcription kit (Biorad, ref. 170–8841), following manufacturer's instruction. 2, 7 ng of cDNA were amplified by quantitative PCR (qPCR) using SYBR Green qPCR SuperMix-UDG with Rox (Thermofisher, ref. 11733046) and 0.1 μM of forward and reverse primers. Levels of transcripts (Ct) were normalized to those of the housekeeping gene GAPDH (ΔCt) and subsequently to the mean ΔCt of the reference sample (ΔΔCt). Results were reported as relative quantities (RQ = $2^{-\Delta\Delta Ct}$). The sequences of forward and reverse primers used for qPCR analyses are listed in Supplementary Table 6.

### RNA sequencing analyses
For RNA-seq, $4 \times 10^4$ cell/cm² cells of WT, shCTRL and shGPC4-1 029 hiPSCs were seeded in triplicate in 12-well plates. After 3 days they were dissociated for RNA extraction as described above. RNA quality was then tested with the instrument 2100 Bioanalyzer Agilent (Purity: OD 260/280: 1.8–2.0; OD 260/230: 2.0–2.2; RIN or RQI value ≥8). 1 μg of RNA per sample at a concentration above 20 ng/μl dissolved in RNase-, DNase- and protease-free molecular grade water was sent to GATC Biotech. The RNA-sequencing was performed by using Illumina HiSeq 2500 technology. The quality of the raw reads was assessed using FastQC v0.11.9 toolkit (Andrews, 2010). Adapters and low-quality reads were trimmed using Trimmomatic v0.39 (Bolger et al., 2014). Paired-end reads were aligned with HiSat2 v2.2.1 (Kim et al., 2015) using default options. Gene counts were quantified using feature Counts v2.0.1 from the Subread v2.0.1 package (Liao et al., 2019). Alignment and gene counts were generated against the GRCh38.p13 (Ensembl release 101). The low expressed genes which did not have more than one count per million reads (1CPM) in at least three samples within each dataset were removed from further analysis. Gene counts were then normalized and used for differential expression testing using DESeq2 v1.28.0[60] (Supplementary Tables 1, 2).

### Western-blots
For western-blots, cells were lysed in 1X EBM lysis buffer in the presence of a protease and phosphatase inhibitors mixture[15]. 20–50 μg of whole cell lysates were resolved on 10% Anderson gels and transferred to PVDF membranes. After blocking with 5% milk in PBS, 0.1% Tween 100, the membranes were incubated overnight at 4 °C with primary antibodies. Primary antibodies used are listed in Supplementary Table 7. The membranes were then washed, incubated with anti-rabbit or anti-mouse IgG-peroxidase (1:4000, Jackson, ref. 211-035-109) at room temperature for 1 hr, and peroxidase activity was visualized with the ECL Plus Kit (Amersham). Full scan blots are provided in the Source Data file.

### Immuno-cytochemical analyses
Cells were fixed in 4% paraformaldehyde (PFA) for 10 min and washed three times with PBS. Permeabilization was done by incubating the cells with 0.3% or 0.1% Triton X100 in PBS for 20 min. Non-specific interactions were blocked for 1 h with blocking solution (3% BSA, Sigma ref. A9647, 2% donkey serum, Abcam ref. ab7475, 0.3% or 0.1% Triton X-100, PBS). Primary antibodies were diluted in the blocking solution and incubated overnight at 4 °C. Primary antibodies used for immunocytochemical analyses are listed in Supplementary Table 8. Secondary antibodies were diluted 1/500 in blocking solution and incubated for 1 h at room temperature. Nuclei were stained with DAPI (1/1000 in PBS) for 5 min. Finally, coverslips were mounted on slides using Fluoromount-G mounting medium. Pictures were acquired with a confocal Zeiss LSM 880 microscope and analyzed through Image J (version 2.3.051) and Zen Blue (version 3.4.9100000) softwares.

### Flow cytometry
Undifferentiated or differentiated hiPSCs were isolated from mono-layer cultures by enzymatic digestion using Accutase. Number of cells for each sample was determined by using Attune NxT Cytometer (Thermo Scientific). 500,000 cells were used for each immunostaining reactions. Fc-Receptors were blocked using anti-CD32 (BD Bioscience, 3D3; 0.5 μg per reaction). Cell surface of undifferentiated hiPSCs and MES differentiated cells were stained for 30 min at 4 °C with anti-human anti-PDGFRa or isotype control following manufacturer recommendation for dilutions. Intracellular staining for EOMES and respective isotype control were performed following manufacturer recommendation (Thermo Scientific, #00-5523-00). Antibodies used

for flow cytometry analyses are listed in Supplementary Table 9. Live/Dead cell discrimination was assayed either using DAPI (MERCK, #D9542) or L/D fixable Blue (Thermo Scientific, #L23105). Samples were analyzed directly upon completion of immunostaining protocol. Data was acquired on BD LSRFortessa 5Laser SORP cytometer (BD Bioscience) using HTS plate reader and analyzed on a standardized analysis matrix using BD FACS DIVA 9.0.1 software. Illustrations were generated on Cytobank 9.4 (Beckman Coulter).

### Quantifications of immunostaining

**Signal quantification and count.** For differentiation and stimulation assays, the number of positive cells showing nuclear localization of transcription factors (LEF1, BRACH and SOX17), downstream effectors (pSMAD2 or pSMAD1), were automatically counted by Image J (version 2.3.051). Median filtering, automatic Huang thresholding and watershed treatment were used to generate a MASK of Nuclei (DAPI) and to count their numbers. For the nuclear marker of interest, background signal was reduced by a Rolling Ball method. Then, Top hat filtering and automatic Otsu thresholding were used to generate a MASK of the nuclear marker and to count positive cells. For cytoplasmic signal (PDGFRa), the number of positive cells were manually counted on Image J (version 2.3.051). Micropatterns experiments were analysed using matlab R21b software.

**Morphological analysis.** To quantify disrupted epithelial areas, ZO1 immunostaining was manually analyzed with Image J (version 2.3.051), and a MASK corresponding to the disrupted area was generated. From this MASK, 4 others concentric MASKs corresponding to four localizations, corresponding to 0 to 30 μm, 30 to 60 μm, 60 to 90 μm or >90 μm away from disrupted areas, were generated and used to analyze the correlation between cells positive for pSMAD1,5, pSMAD2, LEF1 or BRACH (as described in signal quantification and count) and their position relative to the disrupted areas.

**BMPR1A-ZO1 distance analysis.** To quantify BMPR1A-ZO1 distance in the z-axis, BMPR1A and ZO1 z intensity distribution have been manually extracted using Zen Blue (version 3.4.9100000)and the distance in nm between their respective peaks of intensity have been reported.

### Statistical analyses

All statistical analyses were performed with GraphPad Prism version 8 software. All statistical tests used are indicated in the respective figure legends and data were presented as mean ± SEM, unless stated otherwise. Statistics were reported as: ns = not significant, *$p$-value < 0.05, **$p$-value < 0.01, ***$p$-value < 0.001. Precise $p$-value are provided in the source files.

### Reporting summary

Further information on research design is available in the Nature Portfolio Reporting Summary linked to this article.

## Data availability

The data that support this study are available in the source data provided with this paper. For the RNA-seq alignment and gene counts were generated against the GRCh38.p13 (Ensembl release 101; IDs: 2334371 [UID] 8687898 [GenBank] 8765528 [RefSeq]). The RNA-seq data and the read count matrix are available on Zenodo (https://zenodo.org/) with the https://doi.org/10.5281/zenodo.5569383. The RNA-seq data have also been deposited in NCBI's Gene Expression Omnibus and are accessible through GEO Series accession number GSE222186. Source data are provided with this paper.

## Code availability

The workflow allowing to achieve the differential expression analysis of RNA-seq data is available on Zenodo (https://zenodo.org/) with the https://doi.org/10.5281/zenodo.6908427 and is under the BSD-3 license. For the differential RNA-seq analyses, the genes with a log2fc between −0.58 to 0.58 and an adjusted $p$-value (padj) > 0.01 were not considered as significantly differentially expressed.

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

## Acknowledgements

We thank: all members of our labs for helpful discussions and comments; Robert Kelly for critical reading of the manuscript; S. Corti

for contribution during the preparation of RNA for RNAseq studies, E. Bazellières for discussions and reagents. Microscopy was performed at the imaging platform of the IBDM, supported by the ANR through the "Investments for the Future" program (France-BioImaging, ANR-10-INSB-04-01). The authors thank members of the IBDM core facility for microscopy for their technical support. The authors also thank their colleagues (particularly Lillia Hadjem and Hervé Luche), from Immuno-phenotyping service unit at CIPHE (Centre d'Immunophénomique—CIPHE (PHENOMIN), Aix Marseille Université (UMS3367), Inserm (US012), CNRS (UAR3367), Marseille, France) who provided expertise and know-how for the flow cytometry analysis and greatly assisted our project. This project is a part of the research program of the Centre of Excellence DHUNE, which is supported by the French National Plan on Neurode-generative Diseases funded by the French Ministry of national education, higher education and research and the «Investissements d'Avenir» French Government program. The IBDM is affiliated with NeuroMarseille, the AMU neuroscience network, and with NeuroSchool, the AMU grad-uate school in neuroscience supported by the A*MIDEX foundation (AMX-19-IET-004) and the "Investissements d'Avenir" program (nEURo*AMU, ANR-17-EURE-0029 grant). The project leading to this publication has received funding from France 2030, the French Government program managed by the French National Research Agency (ANR-16-CONV-0001) and from Excellence Initiative of Aix-Marseille University—A*MIDEX. This work was supported by fundings from France Parkinson (Convention 096038), Fondation de France (2013_00043173), SATT Sud Est—Accelerator of Technology Transfer, Pre-maturation Program of the CNRS, COEN Pathfinder III (Network of Centres of Excellence in Neurodegeneration; COEN4014) to R.D. D.R. and T.L. were supported by the Ministry of Superior Education and Research Fellowship of France. The funders had no role in study design, data collection and analysis, decision to publish, or preparation of the manuscript.

## Author contributions

T.L., D.R., and R.D. designed the research; F.M. provided input on experiments and data. T.L., D.R., R.D., J.L., conducted experiments; B.S., conducted the 2D gastruloid experiments; T.L., D.R., J.L., B.S., and R.D. analyzed data; T.V. analyzed the RNA-seq analysis; T.L., D.R., J.L., B.S., produced initial images for figures; and T.L., D.R. and R.D. wrote the paper with assistance from all authors. T.L., D.R., J.L., T.V., B.S., F.M. and R.D. reviewed and edited the manuscript. R.D. provided funding.

## Competing interests

The authors declare no competing interests.
