## [Peer Review File · Nature Communications]

Epithelial Disruption Drives Mesendoderm Differentiation in Human Pluripotent Stem Cells by Enabling TGF- β Protein SensingREVIEWER COMMENTS

Reviewer #1 (Remarks to the Author):

The work of Rattier and colleagues demonstrates an important link between epithelial integrity and the ability of early embryonic cells to respond to TGF-beta signalling. Their perturbations can force changes to epithelial polarity in a controllable manner and thereby provide an experimental system to test the importance of apico-basal polarity in responding to BMP signals at the onset of mesodermal specification. The experiments are well performed; however several aspects of the data presentation and main text needs to be improved before it is ready for acceptance.

Major.

1. The rescue of the disrupted TJ organization phenotype by LPA alone is striking. Is the apical-basal polarity phenotype also rescued in these conditions?
2. It will be important to demonstrate a re-localisation of BMP receptors upon Glypican knock-down. At present, this is assumed to be the case from the re-distribution of ZO1 and Na/K ATPase.
3. Multiple experiments are lacking a clear description of n numbers. For example, on how many different repeat experiemnts were the western blot analyses performed? The same is true for many of the immunofluorescence images shown. How many independent experiments performed? And what percentage had the phenotype displayed?

Minor:

- 1) P.g. 7. I am not sure what is meant by "as representative cells carrying the GPC4sh5 and the GPC4sh2 sequences".
- 2) P.g.. "Finally, disruption of TJ organization in GPC4sh hiPSCs occurs in cells plated on different extracellular matrix (ECM) such as Matrigel (mixture of Collagen, Laminin and growth factors), Laminin or Vitronectin (Supplementary Fig. 1c)". What is the significance of this observation?
- 3) Writing style could be improved throughout by ensuring summary sentences are included at the end of each results paragraph.
- 4) "These results were corroborated by western-blot analyses of some epithelial (ZO1, OCLN, E-Cadherin (E-CAD) and β -Catenin (β -cat)) and the mesenchymal (N-Cadherin (N-CAD)) markers". Please include a reference to the appropriate figure panel
- 5) It is not possible to read the gene names in Figure 1E and 2A, could an enlarged version be included in the supplementary material?
- 6) "Consistently, RNAseq analysis revealed that genes regulating fate acquisition such as GSC, BRACH, EOMES, FOXC1, FOXA1 and SOX1 were not expressed neither in WT and CTRLsh hiPSCs nor in GPC4sh5 cells". What figure does this relate to?

Reviewer #2 (Remarks to the Author):

This paper from the Dono lab investigates the effect of epithelial disruption in hiPSCs on pluripotency and differentiation towards mesoderm and endoderm. Using knockdown of GLYPICAN-4 to disrupt epithelial integrity (and also blebbistatin) the authors show that this has no effect on pluripotency, but it promoted the differentiation to mesoderm and endoderm. They show that the reason for this is enhanced and prolonged signaling in response to Activin and BMP. Signaling in response to WNT3 is unaffected.

The work is nicely done, and the data are of high quality. However, the main issue is that the study

lacks novelty. It has been known for some time from the work of Alain Mauviel and Ali Brivanlou and Eric Siggia that the receptors for TGF- β family ligands are confined to the lateral/basal location in an intact epithelial sheet, and are excluded from the apical surface. This means that when the cells are grown on plastic or coverslips, the receptors are only readily accessible at the edges of colonies. These outer cells thus signal first and the signals then can spread within the colony as the cells start to endogenously express the ligands themselves. Therefore, it is not at all surprising that disrupting epithelial integrity would promote accessibility of receptors, which would result in enhanced signaling in response to Activin and BMP. Given that these signals are known to promote mesoderm and endoderm differentiation, it is not surprising that this is enhanced in these conditions.

For the paper to be suitable for publication in Nat Comms, I think that much more mechanism is required to explain why the responses to Activin and BMP are not only enhanced but are prolonged. It is also important for the authors to work out the in vivo relevance of their findings. As it stands, the paper uses a very artificial in vitro system to demonstrate that epithelial integrity constrains the responses to Activin and BMP. Why is this important in vivo? How does the apical/basal location of receptors in vivo influence how differentiation to mesoderm and endoderm is orchestrated.

In addition to these general issues, I have some specific points.

1. In Fig 1g it is a bit difficult to see what is going on, when just the Z axis shown. In the images in Fig 1C there are entire areas where ZO1 staining is missing. What does the Na⁺/K⁺ ATPase staining look like in these regions?
2. The authors show unchanged levels of phosphorylated SMAD2, phosphorylated ERK, phosphorylated AKT and phosphorylated GSK. It would be good to turn off these pathways with inhibitors and thus show that these antibodies are specific.
3. The authors rescue the epithelial integrity with LPA. They need to explain better why this works.
4. For reading out Wnt3A signaling, wouldn't it be better to look at nuclear β -catenin levels?

Reviewer #3 (Remarks to the Author):

In the present manuscript, entitled 'Disruption of Epithelial Integrity Drives Mesendoderm Differentiation in Human Induced Pluripotent Stem Cells by Enabling TGF- β Protein Sensing', Rattier D, Legier T and colleagues identify a crucial role for the morphogen regulator GLYPICAN-4 in preserving the epithelial integrity in 2D human gastruloids and suggest that it secures proper cell fate acquisition by regulating TGF- β signalling activity.

Specifically, the authors compare WT and GLYPICAN-4 KD micropatterned hiPSCs/2D gastruloids, and show that downregulation of GLYPICAN-4 results in tight junctions' defects and eventually formation of disrupted epithelial integrity areas. Immunofluorescence analysis show that the cells located in these areas become competent to perceive BMP4 and ACTIVIN A/NODAL, acquiring endoderm/mesoderm differentiation potential. Furthermore, combining genetic and pharmacological approaches to modulate epithelial integrity, the authors provide solid experimental evidence of their findings.

Overall, this is an interesting study that highlights a mechanism by which epithelial cell integrity controls TGF- β activity during the early phases of the germ layer formation. The data are convincing and very well presented. However, there are important issues that should be addressed to support authors conclusions and the authors need to better contextualise their findings in the literature.

Major points:

- 1. To analyze/quantify the distribution of the different cellular subsets different experimental

approaches should be used alongside Immunofluorescence analysis. Specifically, FACS-based quantification should be included when technically feasible, including for instance in Fig.2f to quantify the percentage of PDGFR α positive cells.

-2. The authors should take into account the difference between Activin and Nodal signaling since Activin and Nodal are not equivalent (Hayes, Kim and Pera, *Stem Cells* 2021 DOI: 10.1002/stem.3383). This is highly relevant in this context; authors used recombinant Activin and conclude that ACTIVIN A/NODAL signals are altered in GPC4 down-regulated cultures (Abstract, page 15 etc). This should be clearly discussed and the manuscript revised accordingly. In this context, it is puzzling to note that the authors ignore a key component of the Nodal pathway that is the obligate Nodal coreceptor Cripto (Hayes, Kim and Pera, *Stem Cells* 2021 DOI: 10.1002/stem.3383; Fiorenzano et al. *Nature Comm* 2016). Expression of Cripto should be analysed alongside that of Nodal at both RNA and protein level, and the relevant references should be cited. Of particular relevance, Nodal and Cripto have been recently shown to be required for the generation of 3D mouse gastruloids (Turner, D.A., et al. *Development* 2017; Cermola F. et al, *Stem Cell Report* 2021) and Activin cannot compensate for the lack of Cripto in gastruloid formation.

3. The authors show that disrupted TJ organization is triggered by GPC4 down-regulation using two different hiPSC lines and properly conclude that the observed phenotype is not limited to only one cell line (page 8). However, this is not sufficient to conclude that disruption of the epithelial integrity is due to GPC4 down-regulation. GPC4 levels should be restored in GPC4sh hiPSCs and it should be demonstrated that this is sufficient to rescue the mutant phenotype.

-4. Page 6. "By performing stimulation assays with BMP4, ACTIVIN A and WNT3A ligands, we show that epithelial integrity regulates the ability of hiPSCs to respond to BMP4 and ACTIVIN A but not WNT3A". Results do not support authors conclusions. Besides the possibility that substrate stiffness might impact on cell response, which is properly discussed, the dose-dependent effects should be considered and analysed. Indeed, the stimulation assays were performed using single doses of the ligands. For instance, Wnt3A was used at high doses (50ng/ml) which may likely mask the need for GPC4. Dose-dependent experiments should be performed to conclude that GPC4 is specific for BMP4 and ACTIVIN A. Furthermore, activation of Wnt/bcatenin should be analyzed by using different experimental approaches including western blot of nuclear beta catenin and/or luciferase assay. Of relevance in this context, authors have previously shown that Glypican4 reduction affects WNT signalling in mouse ESCs (Fico, A. et al. *Stem Cells* 30 1863-1874 2012; cited on page 5). This apparent inconsistency, if any, should be explained and discussed.

-5. Authors should at least discuss their findings on 2Dgastruloids/micropatterned hiPSCs in the context of the large body of literature on 3D gastruloids. This is particularly relevant considering the importance of cell-cell interaction and the structure of the hPSC colony in guiding morphogen sensing and cell fate acquisition processes. This should be clearly indicated and discussed.

Minor points:

- The gene names on the heat maps (Fig. 1e and 2a) are very difficult to read, the size is too small
- page 12, inconsistency between the text and figures (lines 15 and 17: Fig. 2e is Fig. 2d)
- The standard deviation should be reported either in the text or in the figures.
- GLYPICAN-4 KD hiPSCs is a 'unique system'; this statement should be rephrased (introduction and discussion) since authors cannot rule out the possibility that other cell lines may have an equal potential.

Reply to the referees

We thank the three referees for their appreciation of our work and for their useful suggestions for improving the manuscript and strengthening its impact. Below please find a detailed report of the experiments we performed to answer all concerns.

Reviewer #1:

The work of Rattier and colleagues demonstrates an important link between epithelial integrity and the ability of early embryonic cells to respond to TGF-beta signalling. Their perturbations can force changes to epithelial polarity in a controllable manner and thereby provide an experimental system to test the importance of apico-basal polarity in responding to BMP signals at the onset of mesodermal specification. The experiments are well performed; however several aspects of the data presentation and main text needs to be improved before it is ready for acceptance.

Authors' response:

We thank the reviewer for the supportive comments on our manuscript. We have performed additional experiments to answer major and minor comments.

Major.

1. The rescue of the disrupted TJ organization phenotype by LPA alone is striking. Is the apical-basal polarity phenotype also rescued in these conditions?

Authors' response:

To answer this question we have performed immunostaining of CTRLsh and GPC4sh hiPSCs exposed to LPA treatment with antibodies for the proteins ZO1 and NA+/K+ ATPase, which localize at the apical and the basolateral cell membrane, respectively. As control, we performed the same analysis after exposing CTRLsh and GPC4sh hiPSCs to Blebbistatin to examine the effects of disrupting TJ organization on apical-basal cell polarity independently of GPC4. Results reported in the revised version of Figure 5 (Fig. 5b) show that the LPA treatment rescues not only the TJ organization phenotype but also the altered apical-basal polarity in GPC4sh hiPSCs' TJ disrupted areas. In line with these findings, Blebbistatin treatment disrupts both the TJ organization and the apical-basal polarity in CTRLsh and GPC4sh hiPSCs.

Altogether, these results highlight a crosstalk between TJ organization and apical-basal cell polarity. This data has also been included in the revised version of the manuscript at page 15.

2. It will be important to demonstrate a re-localisation of BMP receptors upon Glypican knock-down. At present, this is assumed to be the case from the re-distribution of ZO1 and Na/K ATPase.

Authors' response:

As suggested by the reviewer, we have analyzed the localization of the BMPR1A and ACTR2B, which bind BMP4 and ACTIVIN A, respectively, and are their most expressed receptors in our cells (as from RNA seq analysis). Results reported in the revised Fig. 6 and Supplementary Fig.5 show exposure of BMPR1A and ACTR2B receptors at the apical surface in areas with disrupted epithelial organization (Fig. 6 e-f and Supp Fig. 5e, respectively). This is caused by changes in the ZO1 protein distribution and in the apical-basal cell polarity (see above). Moreover, we performed molecular analysis of BMPR1A, ACTR2B and BMP4 (RT-qPCR and western blot) to examine whether expression levels are affected in GPC4sh hiPSCs. Results showed no significant differences between CTRLsh and GPC4sh hiPSCs (revised Fig. 6g-h and Supplementary Fig. 5f-g). Therefore, disruption of epithelial integrity in hiPSCs with down-regulated GPC4 does not significantly affect BMPR1A, ACTR2B, BMP4 expression but only BMPR1A and ACTR2B accessibility. These results and conclusions are now included in the revised version of the manuscript at page 18-19.

3. Multiple experiments are lacking a clear description of n numbers. For example, on how many different repeat experiments were the western blot analyses performed? The same is true for many of the immunofluorescence images shown. How many independent experiments performed? And what percentage had the phenotype displayed?

Authors' response:

A clear description of different repeats for each experiment is now present in Figure and Supplementary Figure Legends. Briefly, western blots were done in triplicate (see also Source file). Most immunofluorescence images come from experiments

done at least 3 times with the demonstrated phenotype seen in each repeat. Images coming from the second hiPSC line AICS (Supplementary Fig. 1g) and of the rescue experiment of the ZO1 phenotype (Supplementary Fig. 1b and 1c) were done in duplicate for each individual clone. Images coming from E-CAD N-CAD and PDGFR α expression in Fig. 3b, 3c come from duplicate experiments. In general, the disruption of the ZO1 organization in GPC4 mutant cells was seen at least 20 times and the enhanced differentiation efficiency with BRACH and SOX17 at least 10 times. The percentage of the phenotypes is displayed as follows: **1.** Disruption of ZO1 organization is indicated by numbers on the top left corner of panels in Figure 1c (and described in figure legends). Moreover, results are reported as graph in Figure 1d and 1i; **2.** For the differentiation phenotype and stimulation experiments quantifications are reported as graphs near the images and the number of repeats is in figure legends; **3.** For the cell proliferation and cell survival studies quantifications are reported as graphs near the images.

Minor:

1) P.g. 7. I am not sure what is meant by “as representative cells carrying the GPC4sh5 and the GPC4sh2 sequences”.

Authors' response:

As reported previously, we used shRNA-mediated GPC4 targeting to efficiently and stably reduce GPC4 levels in hiPSCs (Corti et al. Stem Cells Transl Med. 2021). By performing studies in HEK cells and hiPSCs, we identified shRNAs efficiently targeting GPC4 and we named them GPC4sh2, GPC4sh4 and GPC4sh5. We transduced the hiPSC line 029 with lentiviruses encoding these selected shRNAs to generate stable clones from each targeting sequence. GPC4sh5-c10 and GPC4sh2-c3 correspond to representative clones carrying the GPC4sh5 and GPC4sh2 sequences, respectively, which we selected to perform all studies reported in this manuscript. We have modified the text to clarify this issue (page 7 of the revised manuscript).

2) P.g.. “Finally, disruption of TJ organization in GPC4sh hiPSCs occurs in cells plated on different extracellular matrix (ECM) such as Matrigel (mixture of Collagen, Laminin and growth factors), Laminin or Vitronectin (Supplementary Fig. 1c)”. What is

the significance of this observation?

Authors' response:

With these studies we started to explore the robustness of the TJ phenotype characterizing the GPC4sh hiPSCs by testing whether it was recapitulated on defined ECM substrates. To this aim, we cultured control and GPC4sh hiPSCs on Laminin and Vitronectin, which are minimal substrates capable of long-term hPSC culture. Our results suggest that disruption of TJ organization in GPC4sh hiPSCs occurs also in the presence of more stringent and defined cultured conditions. This explanation is now included in the text page 8.

3) Writing style could be improved throughout by ensuring summary sentences are included at the end of each results paragraph.

Authors' response:

We have modified the text according to this reviewer suggestion and added summary sentences in paragraphs where they were missing (done in page 13).

4) "These results were corroborated by western-blot analyses of some epithelial (ZO1, OCLN, E-Cadherin (E-CAD) and β -Catenin (β -cat)) and the mesenchymal (N-Cadherin (N-CAD)) markers". Please include a reference to the appropriate figure panel

Authors' response:

We included the correct references for these Figures in the revised manuscripts: Figure 1f (see page 9).

5) It is not possible to read the gene names in Figure 1E and 2A, could an enlarged version be included in the supplementary material?

Authors' response:

We have added an enlarged version in the Supplementary Tables 1 and 2.

6) "Consistently, RNAseq analysis revealed that genes regulating fate acquisition such as GSC, BRACH, EOMES, FOXC1, FOXA1 and SOX1 were not expressed

neither in WT and CTRLsh hiPSCs nor in GPC4sh5 cells". What figure does this relate to?

Authors' response:

Genes not detected in the RNA-seq analysis do not appear in the RNA-seq data file available on Zenodo (<https://zenodo.org/>; DOI 10.5281/zenodo.5569383 (see Data availability)).

Reviewer #2:

This paper from the Dono lab investigates the effect of epithelial disruption in hiPSCs on pluripotency and differentiation towards mesoderm and endoderm. Using knockdown of GLYPICAN-4 to disrupt epithelial integrity (and also blebbistatin) the authors show that this has no effect on pluripotency, but it promoted the differentiation to mesoderm and endoderm. They show that the reason for this is enhanced and prolonged signaling in response to Activin and BMP. Signaling in response to WNT3 is unaffected.

The work is nicely done, and the data are of high quality. However, the main issue is that the study lacks novelty. It has been known for some time from the work of Alain Mauviel and Ali Brivanlou and Eric Siggia that the receptors for TGF- β family ligands are confined to the lateral/basal location in an intact epithelial sheet, and are excluded from the apical surface. This means that when the cells are grown on plastic or coverslips, the receptors are only readily accessible at the edges of colonies. These outer cells thus signal first and the signals then can spread within the colony as the cells start to endogenously express the ligands themselves. Therefore, it is not at all surprising that disrupting epithelial integrity would promote accessibility of receptors, which would result in enhanced signaling in response to Activin and BMP. Given that these signals are known to promote mesoderm and endoderm differentiation, it is not surprising that this is enhanced in these conditions.

For the paper to be suitable for publication in Nat Comms, I think that much more mechanism is required to explain why the responses to Activin and BMP are not only enhanced but are prolonged.

Authors' response:

We thank the reviewer for the supportive comments on our manuscript. To study the cell response to BMP and Activin in the context of an altered TJ organization in hiPSCs, we performed additional experiments. Briefly, we stimulated cells with BMP and Activin in the presence of LPA and Blebbistatin, which were added to the cultures at time point in which CTRLsh and GPC4sh5 hiPSCs activate BMP4 signaling pathway at the same extent (6 hours; Figure 8a). Analysis of pSMAD1,5 positive cells at 12 and 18 hours revealed that LPA significantly reduced the presence of TJ disrupted areas and blocked activation of the BMP4 signaling pathways in both CTRLsh and GPC4sh5 hiPSCs (Fig. 8a-c and Supplementary Fig. 6a). In contrast, the Blebbi treatment triggered TJ disorganization and activation of BMP4 and ACTIVIN A signaling pathways within disrupted areas (Fig. 8a-c and Supplementary Fig. 6a). Similar results were observed when CTRLsh and GPC4sh5 hiPSCs were stimulated with ACTIVIN A in the presence of LPA and Blebbi (Supplementary Fig. 6b-d). These results show that disruption of epithelial integrity maintains activation of TGF- β signaling in the cells located in disrupted areas for a longer time frame. As addition of LPA prevents signal prolongation in GPC4sh5 hiPSCs, we can conclude that the underlying prolongation mechanism is mainly due to TJ organization properties upstream of any intracellular signaling event.

- It is also important for the authors to work out the in vivo relevance of their findings. As it stands, the paper uses a very artificial in vitro system to demonstrate that epithelial integrity constrains the responses to Activin and BMP. Why is this important in vivo?

Authors' response:

We agree with the referee that an in vivo analysis of our findings by using embryos could be desirable. However, these studies were not compatible with the time frame of a revision as it will take 1 to 2 years. Nevertheless, we have decided to take advantage of the 2D gastruloid system also known as micropattern that, thanks to work from Alain Mauviel and Ali Brivanlou and Eric Siggia, has emerged as a suitable in vitro system to study aspects of human gastrulation. In the revised version of the manuscript, Figure 4 shows the micropattern analysis performed on CTRLsh and GPC4sh5 hiPSCs. Both CTRLsh and GPC4sh5 hiPSCs self-organize and

differentiate into a radially organized pattern of ectoderm, mesoderm and endoderm (evidenced by the expression profile of SOX2, BRACH and SOX17, respectively) in response to the gastrulation triggering signal BMP4. Interestingly, in GPC4sh5 hiPSCs the spatial domains of BRACH and SOX17 are expanded at the expense of SOX2 showing increased lineage entry properties of GPC4sh5 hiPSCs in mesendoderm (BRACH positive cells) and SOX17 expressing, thus highlighting an enhanced lineage entry property of GPC4sh5 hiPSCs. Therefore, we think that these 2D gastruloid studies confirm and strengthen findings in our in vitro system. These results are included in the revised version of the manuscript in Figure 4 and page 14, 15. We have also reviewed the in vivo relevance of our studies in the discussion section (page 24, 25).

How does the apical/basal location of receptors in vivo influence how differentiation to mesoderm and endoderm is orchestrated.

Authors' response:

Work from of Zhang et al (Nat Commun 10, 4516, 2019) has shown that in the developing mouse epiblast BMP receptors localize to the basolateral cell domain below the tight junctions where they face a narrow interstitial space between the epiblast and the underlying visceral endoderm. This localization and the presence of TJs make receptors inaccessible to BMP4 ligands secreted in the pre-amniotic cavity. Interestingly, BMP ligands can only access receptors at the posterior epiblast edge due to lack of TJs at the border between the extraembryonic ectoderm, and thus will trigger primitive streak formation at this level. Therefore, the apical/basal location of receptors together with TJ organization are required to restrict BMP signaling perception/action and spatial localization of the primitive streak and the subsequent mesoderm/endoderm differentiation. The relevance of our results to these in vivo findings has now been discussed and is included in the discussion section.

In addition to these general issues, I have some specific points.

1. In Fig 1g it is a bit difficult to see what is going on, when just the Z axis shown. In the images in Fig 1C there are entire areas where ZO1 staining is missing. What does the Na⁺/K⁺ ATPase staining look like in these regions?

Authors' response:

To improve the quality of this image we have included in the Figure 1g a panel showing the XY plane as well, in which TJ disrupted areas found in the GPC4sh5 hiPSCs are indicated by arrowheads. By comparing the panels showing the XY and XZ planes for each cell type we can conclude that the expression levels of Na⁺/K⁺ ATPase proteins are not significantly changed but there are areas in which the basolateral Na⁺/K⁺ ATPase proteins are not below the more apical ZO-1 (TJ) protein (see arrowheads in GPC4sh5 hiPSCs panel).

2. The authors show unchanged levels of phosphorylated SMAD2, phosphorylated ERK, phosphorylated AKT and phosphorylated GSK. It would be good to turn off these pathways with inhibitors and thus show that these antibodies are specific.

Authors' response:

For western blot analysis we have used well established antibodies widely used in papers. Antibody specificity was tested by companies on recombinant proteins and total cell extracts overexpressing or not these proteins. Finally, the specificity of our staining is also illustrated by the Source file in which whole gels are shown.

3. The authors rescue the epithelial integrity with LPA. They need to explain better why this works.

Authors' response:

We decided to perform rescue experiments with LPA because LPA is known to promote TJ formation through the increase of the apical membrane size. These LPA functions are mediated by multiple mechanisms involving G protein-coupled receptors and downstream targets such as PKC and Ca signaling. The scope of using LPA was to encourage cell-cell contact formation with reagents known to regulate apical domain organization in epithelial cells in order to rescue effects caused by GPC4 loss. To get some insight into whether components of LPA signaling were impacted by GPC4 loss, we analysed the phosphorylation levels of PKC lambda and zeta in control and GPC4sh hiPSCs and found no differences. Given the complexity of LPA signaling mechanisms determining the net contribution

of each component would require a detailed molecular analysis with complementary approaches, which we consider out of the main scope of our studies.

4. For reading out Wnt3A signaling, wouldn't it be better to look at nuclear β -catenin levels?

Authors' response:

We agree with the referee and we have performed an analysis to look at the nuclear β -catenin levels. Our analysis revealed a nuclear localization of β -catenin following WNT3A stimulation in both control and GPC4sh hiPSCs. However, due to simultaneous nuclear and cytoplasmic distribution of β -catenin in WNT3A stimulated cells and the poor quality of the antibodies available, it is complex to perform a robust quantification of the nuclear β -catenin without the interference of its cytoplasmic/membrane expression. Therefore, we have prepared a figure for the referee to show the β -catenin cell distribution following WNT3A stimulation in both control and GPC4sh hiPSCs (Figure for Reviewer). Instead, we prefer to focus on the analysis of LEF1 protein distribution as a readout of WNT3A signaling in the text. However, we would be happy to add this figure to the included supplementary figures at the reviewer's discretion.

Reviewer #3 (Remarks to the Author):

In the present manuscript, entitled 'Disruption of Epithelial Integrity Drives Mesendoderm Differentiation in Human Induced Pluripotent Stem Cells by Enabling TGF- β Protein Sensing', Rattier D, Legier T and colleagues identify a crucial role for the morphogen regulator GLYPICAN-4 in preserving the epithelial integrity in 2D human gastruloids and suggest that it secures proper cell fate acquisition by regulating TGF- β signalling activity.

Specifically, the authors compare WT and GLYPICAN-4 KD micropatterned hiPSCs/2D gastruloids, and show that downregulation of GLYPICAN-4 results in tight junctions' defects and eventually formation of disrupted epithelial integrity areas. Immunofluorescence analysis show that the cells located in these areas become competent to perceive BMP4 and ACTIVIN A/NODAL, acquiring endoderm/mesoderm differentiation potential. Furthermore, combining genetic and

pharmacological approaches to modulate epithelial integrity, the authors provide solid experimental evidence of their findings.

Overall, this is an interesting study that highlights a mechanism by which epithelial cell integrity controls TGF- β activity during the early phases of the germ layer formation. The data are convincing and very well presented. However, there are important issues that should be addressed to support authors conclusions and the authors need to better contextualise their findings in the literature.

Authors' response:

We thank the reviewer for the supportive comments on our manuscript. We have performed additional experiments and modified the text to answer major and minor points. We would like to specify better that in the previous version of the manuscript we performed studies on hiPSCs cultured in classical culture conditions without employing the micropatterned hiPSCs/2D gastruloid technique. We have applied this technique of the 2D gastruloids in this revised version of the manuscript to address specific questions raised by reviewers.

Major points:

-1. To analyze/quantify the distribution of the different cellular subsets different experimental approaches should be used alongside Immunofluorescence analysis. Specifically, FACS-based quantification should be included when technically feasible, including for instance in Fig.2f to quantify the percentage of PDGFRa positive cells.

Authors' response:

We agree with the reviewer's comment and we have performed FACS analysis of several proteins of interest. We have added the FACS analysis of PDGFRa and EOMES positive cells. We have also analyzed by FACS the number of positive cells for other proteins such as BRACH, SOX17 and FOXA2 but we did not succeed to get good quality and reproducible results with the commercial antibodies we used.

-2. The authors should take into account the difference between Activin and Nodal signaling since Activin and Nodal are not equivalent (Hayes, Kim and Pera, Stem Cells 2021 DOI: 10.1002/stem.3383). This is highly relevant in this context; authors

used recombinant Activin and conclude that ACTIVIN A/NODAL signals are altered in GPC4 down-regulated cultures (Abstract, page 15 etc). This should be clearly discussed and the manuscript revised accordingly. In this context, it is puzzling to note that the authors ignore a key component of the Nodal pathway that is the obligate Nodal coreceptor Cripto (Hayes, Kim and Pera, Stem Cells 2021 DOI: 10.1002/stem.3383; Fiorenzano et al. Nature Comm 2016). Expression of Cripto should be analysed alongside that of Nodal at both RNA and protein level, and the relevant references should be cited. Of particular relevance, Nodal and Cripto have been recently shown to be required for the generation of 3D mouse gastruloids (Turner, D.A., et al. Development 2017; Cermola F. et al, Stem Cell Report 2021) and Activin cannot compensate for the lack of Cripto in gastruloid formation.

Authors' response:

We thank the referee for pointing out this issue of NODAL and ACTIVIN A. We have removed the use of "ACTIVIN A/NODAL" and used NODAL only in the introduction when describing the embryonic processes (page 3) and use ACTIVIN for description of our experiments. As specified in the abstract and in page 12 we have used ACTIVIN A as a surrogate, which is commonly used in the literature to activate the Nodal signaling pathway. The relevant references, including those suggested by the reviewer, have been added in the revised version of the manuscript. Finally, we have performed a molecular analysis of NODAL and CRIPTO and did not find significant differences in their expression between CTRLsh and GPC4sh hiPSCs. Whereas we succeeded to perform analysis at both at RNA and protein levels, for NODAL we did not get results of sufficient quality to accurately report the expression levels of NODAL proteins in the different samples (antibody used: Nodal (A-9), SantaCruz, sc-377508). Therefore, we have only included an RT-PCR analysis of *NODAL* transcripts. These results are reported in Supplementary Fig. 5f-g and describe in page 19.

3. The authors show that disrupted TJ organization is triggered by GPC4 down-regulation using two different hiPSC lines and properly conclude that the observed phenotype is not limited to only one cell line (page 8). However, this is not sufficient to conclude that disruption of the epithelial integrity is due to GPC4 down-regulation.

GPC4 levels should be restored in GPC4sh hiPSCs and it should be demonstrated that this is sufficient to rescue the mutant phenotype.

Authors' response:

As suggested by the reviewer, we have done rescue experiments to restore GPC4 levels in GPC4sh hiPSCs. For these studies we used plasmids expressing mouse GPC4, which is not targeted by the hGPC4 shRNA, and plasmids expressing the human GPC4. Unfortunately, transfection experiments done with mouse GPC4 did not generate any GPC4sh hiPSC clone expressing mouse GPC4, although are capable to give rise to transient mouse GPC4 expression. Instead, transfection with human GPC4 produced clones in which GPC4 levels were restored at different extents (GPC4sh5-c10 + hGPC4 cl1 and GPC4sh5-c10 + hGPC4 cl2: around 200% and 300% of the GPC4sh5-c10, respectively). Analysis of ZO1 distributions in these clones revealed a correlation between GPC4 levels in cells and the rescue of epithelial disruption with a higher efficiency when the GPC4 level reaches that of CTRLsh hiPSCs (as for the GPC4sh5-c10 + hGPC4 cl2). Results are reported in Supplementary Fig. 1b-c and described at page 8 and figure legend of Supplementary Fig. 1.

-4. Page 6. "By performing stimulation assays with BMP4, ACTIVIN A and WNT3A ligands, we show that epithelial integrity regulates the ability of hiPSCs to respond to BMP4 and ACTIVIN A but not WNT3A". Results do not support authors conclusions. Besides the possibility that substrate stiffness might impact on cell response, which is properly discussed, the dose-dependent effects should be considered and analysed. Indeed, the stimulation assays were performed using single doses of the ligands. For instance, Wnt3A was used at high doses (50ng/ml) which may likely mask the need for GPC4. Dose-dependent experiments should be performed to conclude that GPC4 is specific for BMP4 and ACTIVIN A. Furthermore, activation of Wnt/bcatenin should be analyzed by using different experimental approaches including western blot of nuclear beta catenin and/or luciferase assay. Of relevance in this context, authors have previously shown that Glypican4 reduction affects WNT signalling in mouse ESCs (Fico, A. et al. Stem Cells 30 1863-1874 2012; cited on page 5). This apparent inconsistency, if any, should be explained and discussed.

Authors' response:

Following the reviewer's suggestion, we performed a dose-dependent experiment to analyze the WNT signaling response in more detail. Results reported in Figure 7d-f show no significant differences in LEF1 expression and distribution at any dosage ranging from 5ng (low activation) to 50ng (full activation) of WNT3A per ml. Moreover, the distribution of LEF1 positive cells does not correlate with areas of disrupted epithelial integrity, thus showing that epithelial integrity does not control the hiPSC response to WNT3A. In addition, we have analyzed β -catenin by western blot and immunostaining. The western blot results in Fig. 1f show no significant differences between CTRL and GPC4sh hiPSC. Concerning the immunocytochemistry of β -catenin our analysis revealed a nuclear localization of β -catenin following WNT3A stimulation in both control and GPC4sh hiPSCs. However, due to simultaneous nuclear and cytoplasmic distribution of β -catenin in WNT3A stimulated cells and the poor quality of the antibodies available, it is complex to perform a robust quantification of the nuclear β -catenin without the interference of its cytoplasmic/membrane expression. Therefore, we have prepared a figure for the reviewer to show the β -catenin distribution in cells following WNT3A stimulation in both control and GPC4sh hiPSCs (Figure for reviewer). We prefer to focus on the analysis of LEF1 protein distribution as a read out of WNT3A signaling in the text. However, we would be happy to add this figure to the included supplementary figures at the reviewer's discretion.

5. Authors should at least discuss their findings on 2Dgastruloids/micropatterned hiPSCs in the context of the large body of literature on 3D gastruloids. This is particularly relevant considering the importance of cell-cell interaction and the structure of the hPSC colony in guiding morphogen sensing and cell fate acquisition processes. This should be clearly indicated and discussed.

Authors' response:

The relevance of our findings in the context of 3D gastruloids is now discussed in page 24.

Minor points:

- The gene names on the heat maps (Fig. 1e and 2a) are very difficult to read, the size is too small

Authors' response:

We apologize for this inconvenient. We have added an enlarged version in Supplementary Tables 1 and 2.

- page 12, inconsistency between the text and figures (lines 15 and 17: Fig. 2e is Fig. 2d)

Authors' response:

We thank the reviewer. This mistake is corrected in the revised version

- The standard deviation should be reported either in the text or in the figures.

Authors' response:

In the revised submission all graphs are either: box plots displaying all data points; box & whisker/violin plots displaying all quartiles; or line plots displaying either the median & interquartile range or the mean +/- SD/SEM. Moreover, data from graphs have been added to the source data file.

- GLYPICAN-4 KD hiPSCs is a 'unique system'; this statement should be rephrased (introduction and discussion) since authors cannot rule out the possibility that other cell lines may have an equal potential.

Authors' response:

We have rephrased this statement, as GLYPICAN-4 KD hiPSCs are a 'relevant system'.

REVIEWER COMMENTS

Reviewer #1 (Remarks to the Author):

The authors have satisfactorily addressed all my previous concerns. Significant effort has gone into this revised article, that I now feel is ready for publication.

Reviewer #2 (Remarks to the Author):

The authors have performed some major revisions on their paper and it is certainly improved in many aspects. However, in my view there are still some outstanding issues.

The authors still have not explained the mechanism whereby BMP and Activin signaling is sustained as well as enhanced as a result of GPC4 knockdown. They now conclude that it is due to TJ organization properties, but this doesn't explain the mechanism. This still needs to be addressed.

In the new experiment that the authors present in Figure 8 – why do the control cells in Figure 8a not have nice regular ZO1 staining as the cells do in Figure 6a? This is a concern, because in Figure 8a the control and GPC4sh cells look very similar with respect to ZO1 staining, whilst in other figures the controls and GPC4sh cells look very different.

I had asked that the authors test the specificity of the antibodies against phosphorylated SMAD2, phosphorylated ERK, phosphorylated AKT and phosphorylated GSK by turning off the pathways leading to these phosphorylations and showing that the signal is inhibited. They have not done this, but it is a crucial control that needs to be done.

It is still not explained why GPC4 knockdown causes the disruption of epithelial integrity. This needs to be addressed.

Minor point.

The authors use the term TGF- β signaling generally at the end of the introduction and on page 20. This is confusing as it sounds as though they are using TGF- β as the ligand, when in fact they are talking about BMP4 and Activin.

Reviewer #3 (Remarks to the Author):

The authors have addressed the main concerns and improved the quality of the manuscript. I recommend publication of the manuscript in Nature Communications.

Minor:

The role of Cripto as Nodal coreceptor is not properly cited (page 19, line 432; reference 48 and 49).
Ref. 48: Cadigan, K.M. & Waterman, M.L. TCF/LEFs and Wnt signaling in the nucleus. Cold Spring Harb Perspect Biol 4 (2012).

Ref. 49: Estaras, C., Benner, 1030 C. & Jones, K.A. SMADs and YAP compete to control elongation of beta-catenin:LEF-1-recruited RNAPII during hESC differentiation. Mol Cell 58, 780-793 (2015).

Reply to the reviewers

We were pleased to see that all reviewers appreciated the major revision we have performed to improve the manuscript quality. Moreover, we are pleased to see that Reviewers 1 and 3 were fully satisfied from this additional work and recommend publication in Nature Communications. However, it was unfortunate to see that Reviewer 2 has still some concerns about this revised version.

Below please find a detailed report of the experiments we performed to answer all concerns.

Reviewer #1 (Remarks to the Author):

The authors have satisfactorily addressed all my previous concerns. Significant effort has gone into this revised article, that I now feel is ready for publication.

Authors' response:

We thank the reviewer for appreciating our revised manuscript.

Reviewer #2 (Remarks to the Author):

The authors have performed some major revisions on their paper and it is certainly improved in many aspects. However, in my view there are still some outstanding issues.

Authors' response:

We thank the reviewer for appreciating our revised manuscript. Below please find a detailed report of the experiments we performed to answer all concerns.

The authors still have not explained the mechanism whereby BMP and Activin signaling is sustained as well as enhanced as a result of GPC4 knockdown. They now conclude that it is due to TJ organization properties, but this doesn't explain the mechanism. This still needs to be addressed.

Authors' response:

To address this issue, we have focused on the biological mechanism whereby the responses to BMP and ACTIVIN are not only enhanced but also sustained as a consequence of GPC4 knockdown in hiPSCs. The scope of the previous revised versions of Figure 8 and Supplementary 6 was to illustrate that this sustained activation of BMP and ACTIVIN signaling in GPC4sh hiPSCs correlated with an altered capability of cells to establish TJs overtime (GPC4sh hiPSCs, CTRLsh+Blebbi and GPC4sh+LPA). However, Reviewer 2's comments (see also the next question) made us realize that the presented results were not sufficient to illustrate this biological mechanism, nor were they clearly described. Therefore, we performed additional studies to examine thoroughly and quantify the temporal dynamics of TJ formation in GPC4sh and CTRLsh hiPSCs as well as its correlation with activation of BMP and ACTIVIN A signaling. These new results are reported in the revised version of Figure 8, Supplementary 6, and of the manuscript text at page 20 to 23. Briefly, cells were dissociated and seeded with ROCK inhibitor for 24 hours. Then, ROCK inhibitor was removed and cells left to recover for 24 hours prior to BMP4 and ACTIVIN A stimulation. Cells were fixed at 0, 2, 6, 9, 12 and 18 hours following stimulation, and immunostained with ZO1 and pSMAD1,5 and pSMAD2 to examine their ability to activate the BMP4 and ACTIVIN A signaling pathways, respectively. As shown in Figure 8a and Supplementary 6, analysis of stimulated control hiPSCs at 9/12 hours revealed recovery of TJs as well as predominant restriction of pSMAD1,5 and pSMAD2 positive cells to the few areas with disrupted ZO1 organization. In contrast, recovery of TJ organization over time occurs to a lower extent in GPC4sh cells after seeding, as highlighted by the presence of larger areas of disrupted ZO1 organization in cells stimulated for 9/12 hours. Interestingly, cells within these disrupted areas remain positive for pSMAD1,5 and pSMAD2 staining showing a sustained activation of BMP and ACTIVIN signaling over time. These qualitative results are strengthened by our quantitative analysis in Figure 8b showing a clear correlation between protracted TJ disorganization and sustained pSMAD1,5 expression.

As reported in the previous revised version of the manuscript and of Figure 8, we also examined pSMAD1,5, pSMAD2, and ZO1 expression in control and shGPC4 stimulated cells in situations in which TJ formation is impaired or promoted by drugs (stimulated cells for 12/18 hours and treated with Blebbi and LPA 6h after stimulation

onset) and clearly observed a delay in BMP4 and ACTIVIN A pathway shutdown when TJ formation is impaired (CTRLsh+Blebbi and GPC4sh±Blebbi) whereas restoring TJ formation with LPA rescues the dynamics of pathway shutdown, showing causality (CTRLsh±LPA and GPC4sh+LPA). In our opinion these overall new findings explain why the responses to ACTIVIN and BMP are prolonged in GPC4sh hiPSCs by showing that the underlying mechanism involves a delay in TJ formation.

In the new experiment that the authors present in Figure 8 – why do the control cells in Figure 8a not have nice regular ZO1 staining as the cells do in Figure 6a? This is a concern, because in Figure 8a the control and GPC4sh cells look very similar with respect to ZO1 staining, whilst in other figures the controls and GPC4sh cells look very different.

Authors' response:

As above described, the experiment of Figure 8a is a time course analysis of TJ formations. In brief, cells were first dissociated and seeded with ROCK inhibitor for 24 hours, then ROCK inhibitor was removed and cells left to recover for 24 hours prior to BMP stimulation. Following stimulation, cells were fixed at 0, 2, 6, 9, 12 and 18 hours and immunostained with ZO1 (and pSMAD1,5). At 0 and 2 hours post-stimulation control and GPC4sh cells look very similar with respect to ZO1 staining, as they have not yet recovered TJs lost during the dissociation procedure. Significant differences become apparent from 6 to 12 hours in which the ZO1 organization becomes more pronounced in control versus GPC4sh cells, as time progresses. These overall results illustrate that the differences in ZO1 staining between control and shGPC4 hiPSCs become apparent over time post seeding (see above).

In other figures of the manuscript analysis has been done at the time point in which differences in the ZO1 staining between control and GPC4sh cells is already apparent. This is now specified for each experiment in Material and Methods.

I had asked that the authors test the specificity of the antibodies against phosphorylated SMAD2, phosphorylated ERK, phosphorylated AKT and

phosphorylated GSK by turning off the pathways leading to these phosphorylations and showing that the signal is inhibited.

They have not done this, but it is a crucial control that needs to be done.

Authors' response:

We apologize for not being able to perform these experiments in the previous revised version of the manuscript. We have now analyzed specificity of the antibodies for the phosphorylated forms of SMAD2, ERK, AKT and GSK3 by using specific chemical reagents preventing their phosphorylation. Results are reported in Supplementary Figure 2 and described in its Figure legend.

It is still not explained why GPC4 knockdown causes the disruption of epithelial integrity. This needs to be addressed.

Authors' response:

By reading the comments of Reviewer 2 in this 2nd round of revision it is our understanding that they asks now to go deeper with this analysis and link the phenotype of TJ organization to the molecular mechanisms triggered by GPC4 knockdown. We agree with the reviewer that deciphering the molecular mechanism by which GPC4 knockdown causes this phenotype would be desirable. However, this new request represents a very large amount of work and time involving multi-omics approaches such as proteomics and functional validations, which we consider out of the main scope of this study and the topic of a second paper. Nonetheless, we have analyzed our RNA-sequencing data (Figures 1e, 2a, extended data file), and identified genes differentially expressed in GPC4sh hiPSCs versus control such as *MMP9*, *SEMA6D*, *DGKK*, *CDH6*, and *RASIP1*, which could potentially impact on the regulation of epithelial organization, as these proteins are implicated in several biological processes such as extracellular matrix remodeling, LPA metabolism, cell adhesion and GTPase regulation (see complete list in the figure for reviewer) [Al-Sadi R et al. PLoS One. 2021 Apr 7;16(4):e0249544. doi: 10.1371/journal.pone.0249544; Treps L et al. Tissue Barriers. 2013 Jan 1;1(1):e23272. doi: 10.4161/tisb.23272; Arisz, Steven & Munnik, Teun. Journal of lipid research 2011: 52. 2012-20. Doi:10.1194/jlr.M016873; de Kreug BJ et al. eLife

2016: 5:e11394. doi.org/10.7554/eLife.11394]. The differential expression of these genes in GPC4sh hiPSCs has been validated by RT-qPCR, and we have prepared a figure for the reviewer reporting these data. Also, we initiated functional studies for RASIP1 by focusing on RASIP1 interaction with GTPase proteins. Unfortunately, these studies did not provide significant differences between control and GPC4sh hiPSCs. As these genes are implicated in multiple biological processes, we realized that additional omics approaches such as proteomics and IP-mass spectrometry are needed to understand why these genes are differentially expressed and whether they are linked to the TJ phenotype of GPC4sh hiPSCs. Finally, we would like to point out that GPC4 is known to function as modulator of different signaling pathways and does not have “its own downstream signaling cascade”. In our opinion, this situation makes particularly challenging to link these changes in RNA levels to distinct GPC4 protein function, which could instead be addressed through large scale analyses as discussed above. Therefore, these and other molecular studies constitute a considerable amount of investment and analysis, which we consider the scope of a second manuscript.

Minor point. The authors use the term TGF- β signaling generally at the end of the introduction and on page 20. This is confusing as it sounds as though they are using TGF- β as the ligand, when in fact they are talking about BMP4 and Activin.

Authors' response:

We have replaced TGF- β signaling with BMP4 and ACTIVIN A signaling at the end of introduction and on page 20.

Reviewer #3 (Remarks to the Author):

The authors have addressed the main concerns and improved the quality of the manuscript. I recommend publication of the manuscript in Nature Communications.

Authors' response:

We thank the reviewer for appreciating our revised manuscript.

Minor:

The role of Cripto as Nodal coreceptor is not properly cited (page 19, line 432; reference 48 and 49). Ref. 48: Cadigan, K.M. & Waterman, M.L. TCF/LEFs and Wnt signaling in the nucleus. Cold Spring Harb Perspect Biol 4 (2012). Ref. 49: Estaras, C., Benner, 1030 C. & Jones, K.A. SMADs and YAP compete to control elongation of beta-catenin:LEF-1-recruited RNAPII during hESC differentiation. Mol Cell 58, 780-793 (2015).

Authors' response:

We apologize for this mistake, which occurred during the formatting process. The text has now been modified to cite properly the role of Cripto as a Nodal coreceptor (page 19 and pages 44 and 45).

Figure for Reviewer

a

found to be down or up regulated ?	gene	RNA seq		qPCR validation	linked to	Short gene description
		log2 foldchange	pvalue			
-	CHCHD2	-1.65	1.75E-101	significant	mitochondria	Mitochondria associated
-	DNAJC15	-2.98	5.38E-27	ND	mitochondria	Linked to mitochondrial hyperpolarization and ATP generation
-	FEZF1	-1.14	9.55E-74	significant		Regulation of the olfactory bulb development and the interneurons
-	HTR1A	-0.63	7.50E-12	significant	calcium	Inhibition of adenylate cyclase activity and activation a phosphatidylinositol-calcium second messenger
-	PLD6	-0.94	1.16E-04	not significant	mitoC / LPA	Acting as a cardiolipin hydrolase to generate phosphatidic acid
-	SCN4B	-0.72	2.31E-06	ND	ions transport	Causing negative shifts in the voltage dependence of activation of certain alpha sodium channels
-	DGKK	-0.70	6.15E-23	significant	LPA	Phosphorylates diacylglycerol (DAG) to generate phosphatidic acid (PA)
-	ADAMTS5	-0.60	2.15E-06	ND	ECM	Extracellular matrix (ECM) degrading enzyme
-	CDH6	-0.61	1.80E-07	significant	adhesion	Cell adhesion proteins
-	SEMA6D	-0.83	3.88E-11	ND		Maintenance and remodeling of neuronal connections
-	MAP2	-0.86	7.79E-20	ND	trafficking	Stabilization of the microtubules against depolymerization
+	BNIP3	1.26	3.22E-47	not significant	calcium	Possible role in repartitioning calcium between the two major intracellular calcium stores
+	COL6A3	0.78	1.58E-35	significant	ECM	ECM component
+	MDGA2	0.94	6.42E-13	significant	adhesion	May be involved in cell-cell interactions
+	MMP9	0.95	2.43E-14	ND	ECM	Role in local proteolysis of the extracellular matrix
+	BGN	0.68	1.44E-04	ND	ECM	May be involved in collagen fiber assembly
+	CALCR	0.63	2.54E-06	ND		Receptor for calcitonin gene-related peptide , endocytosis modulator
+	PCDH10	0.69	1.40E-11	ND	adhesion	Potential calcium-dependent cell-adhesion protein
+	PCDHA3	1.11	1.61E-26	ND	adhesion	Potential calcium-dependent cell-adhesion protein
+	PCDHB3	1.00	1.20E-44	ND	adhesion	Potential calcium-dependent cell-adhesion protein
+	PCDHGA11	1.09	2.55E-35	ND	adhesion	Potential calcium-dependent cell-adhesion protein
+	PCDHGB5	0.62	1.22E-06	ND	adhesion	Potential calcium-dependent cell-adhesion protein
+	PEX5L	0.93	8.85E-09	ND	ions transport	Accessory subunit of hyperpolarization-activated cyclic nucleotide-gated (HCN) channels
+	RASIP1	0.78	3.79E-09	significant	GTPase	Implication as a critical and vascular-specific regulator of GTPase signaling, cell architecture, and adhesion
+	SULF1	0.86	2.37E-20	ND	GPC	Arylsulfatase activity and highly specific endoglucosamine-6-sulfatase activity
+	ARRB1	0.69	3.35E-18	ND	GTPase	Members of arrestin/beta-arrestin protein family thought to modulate G-protein-coupled receptors
+	EDNRB	0.76	1.39E-17	ND	GTPase / calcium	G protein-coupled receptor activating a phosphatidylinositol-calcium second messenger system
+	ARHGGEF5_1	0.96	7.97E-05	ND	GTPase	Rho Guanine Nucleotide Exchange Factor

b

Figure Legend:

(a) List of selected genes differentially expressed in GPC4sh versus CTRLsh hiPSCs. Fold differences and P values from RNAseq analysis are shown as well as the results of the RTqPCR validation done on a subset of gene (ND= not done). A short description of gene function is also included. (b) Graphs showing the RTqPCR validation above reported. Box plots represent the median with interquartile range, the whiskers indicate min and max values, n = from 6 to 8. Statistical analysis: paired t-test. P values: (***) < 0.001, (**) < 0.01, (*) < 0.05.

REVIEWERS' COMMENTS

Reviewer #2 (Remarks to the Author):

The authors have done a good job in responding to my comments with new data. I have a question, though, about the data in Supplementary Figure 6a. The Activin induction is shown at 0, 2, 12 and 18 hours with PSmad2 staining used as a readout for Activin signaling. The 0 time point however has equally high levels of PSmad2 staining as the 2 h and 12 h time points after Activin stimulation. Why is this?

Reply to the Reviewer2

Reviewer#2 (Remarks to the Author):

The authors have done a good job in responding to my comments with new data. I have a question, though, about the data in Supplementary Figure 6a. The Activin induction is shown at 0, 2 12 and 18 hours with PSmad2 staining used as a readout for Activin signaling. The 0 time point however has equally high levels of PSmad2 staining as the 2 h and 12 h time points after Activin stimulation. Why is this?

We were pleased to see that Reviewer 2 appreciated the revision we have performed to address concerns, and is fully satisfied from this additional work. Concerning the question about why *“The 0 time point however has equally high levels of PSmad2 staining at 2 h and 12 h time points after Activin stimulation”*, at t=0 (before stimulation) pSMAD2 is active due to the presence of TGFb1 in mTeSR1 medium in which the cells are grown and stimulated. Indeed, TGFb1 activates pSMAD2 together with ACTIVIN A. Thus, pSMAD2 is already activated at d0. While performing the different biological replicates, we could observe an increase of intensity in pSMAD2 levels between 0 and 2h after ACTIVIN A stimulation. Images are a bit saturated so this may not be obvious. However, this increase was not robust enough to reach a clear-cut conclusion. We think that this is most likely due to the to the adaptive nature of the ACTIVIN/NODAL pathway in hESC (as described by Yoney et al Elife 2018 <https://doi.org/10.7554/eLife.38279>).